# Coccolith clumped isotopes reveal modest rather than extreme northern high latitude amplification during the Miocene

Luz María Mejía [1,5] ✉, Stefano M. Bernasconi[1], Alvaro Fernandez[2], Hongrui Zhang[1,6], José Guitián [1,7], Madalina Jaggi[1], Victoria E. Taylor [3], Alberto Perez-Huerta[4] & Heather Stoll [1]

Accurate predictions of the future climate response to $CO_2$ depend on the ability of climate models to simulate past analog warmer climates, like the Miocene. However, one key unresolved issue in paleoclimate modeling is reproducing the pronounced high-latitude warmth and relatively flat latitudinal temperature gradients inferred from proxy records. Here, we use clumped isotope thermometry—a method that sidesteps limitations of conventional proxies—on pure coccolith calcite from a high-latitude North Atlantic site, extending from the Mid Miocene to the Quaternary. Coccolith-derived clumped isotope temperatures are on average ~9°C lower than alkenone estimates, representing the first proxy dataset to align with Miocene model outputs and calling into question the prevailing paradigm of pronounced high latitude amplification. This record highlights the need to continuously reevaluate proxy interpretations to achieve both reliable trends and absolute temperature values, while providing a more optimistic perspective of future high latitude climate response to $CO_2$ emissions.

Temperature indicators, such as foraminiferal $\delta^{18}O$ and Mg/Ca, archaeal tetraether index ($TEX_{86}$) and alkenone unsaturation index ($U^k_{37}$) generally show a consistent global cooling trend over the Cenozoic (e.g., refs. 1–3). Such estimates have been used to test whether climate models used to predict future climate can accurately simulate earth's climate response under high $CO_2$ atmospheric concentrations ($pCO_2$), and other boundary conditions which differ from those of the observational period used to tune models. To date, one of the largest model-data discrepancies occur in the simulation of high latitude warmth, especially for the Miocene[4–6], because proxy data imply strong high latitude amplification and flattening of the latitudinal thermal gradient during warmer climate states (e.g., refs. 1,7,8). It is therefore unclear whether climate models are missing key physical processes, or if the

validity of reconstructed absolute temperature estimates and/or interpretation of well-established proxies needs to be re-examined, both of which can hamper accurate predictions of future climate. From all potential past analogs to future climate conditions, the Miocene (~5.33–23.03 million years ago, Ma) is perhaps currently the most important, since we have already surpassed ($pCO_2$ today: 427 ppm) the $pCO_2$ levels of the younger Pliocene (~2.58–5.33 Ma; <400 ppm)[5,9], and the modern continental configuration is significantly different to that of the older Eocene (~33.9–55.8 Ma; >800 ppm)[5,9]. The Miocene corresponds to middle, more realistic, future emission scenarios (RCP 4.5–6.0)[5], with estimated $pCO_2$ concentrations of ~400–600 ppm[5,9] and a more similar continental configuration to the modern. Therefore, to improve our understanding of the thermal response of high latitudes to $CO_2$ forcing,

[1]Geological Institute, ETH, Zürich, Switzerland. [2]Instituto Andaluz de Ciencias de la Tierra (CSIC), Granada, Spain. [3]Department of Earth Science and Bjerknes Centre for Climate Research, University of Bergen, Bergen, Norway. [4]Department of Geological Sciences, University of Alabama, Tuscaloosa, AL, USA. [5]Present address: MARUM, University of Bremen, Bremen, Germany. [6]Present address: Tongji University, Shanghai, China. [7]Present address: Department of Oceanography, Instituto de Investigacións Mariñas (CSIC), Vigo, Spain. ✉e-mail: lmejia@marum.de

robust reconstructions of high latitude surface ocean temperatures from the Miocene are necessary.

Clumped isotope ($\Delta_{47}$) thermometry is a technique that estimates calcification temperatures based on the excess abundance of $^{13}$C-$^{18}$O bonds, which are more stable at lower temperatures, compared to their abundance if the rare isotopes $^{13}$C and $^{18}$O were stochastically distributed among all isotopologues[10]. The application of $\Delta_{47}$ to reconstruct temperatures has the advantage of being independent of seawater chemistry, in contrast to foraminiferal Mg/Ca and $\delta^{18}$O[11]. Moreover, widely-used biomarkers, such as TEX$_{86}$ and U$^{k'}_{37}$, are uniquely based on empirical correlations to temperature, and the mechanism(s) driving these correlations are not well known. In contrast, the relationship between $\Delta_{47}$ and temperature is well understood and grounded in thermodynamics. Since most of the surface ocean temperature reconstructions from the Miocene are based on U$^{k'}_{37}$[4], new records based on $\Delta_{47}$ thermometry during this time allow us to improve the reliability of absolute temperature estimates. Recent improvements in the precision, methods, and calibrations, and reduction in sample size requirements of $\Delta_{47}$ thermometry have made it useful for paleoceanographic applications (e.g., refs. 11,12).

Despite being geographically widespread since the Mesozoic and ensuring a euphotic ocean signal due to their reliance on light, the application of $\Delta_{47}$ to calcite produced by coccolithophores has received attention by the community only recently[13–18]. Two studies on cultured coccolithophores showed that despite the large vital effects in $\delta^{18}$O and $\delta^{13}$C, the relationship between coccolith $\Delta_{47}$ and temperature appears to be consistent across different species[14,18]. Altogether, available data suggests that the application of $\Delta_{47}$ to coccolith samples of mixed species is a reliable indicator of coccolithophores' calcification temperature, which in well-mixed waters, like at high latitudes, likely reflect integrated mixed-layer temperatures during the production season[17,19].

Here we applied $\Delta_{47}$ thermometry to exceptionally pure (>90%) downcore coccolith calcite from ODP site 982 in the North Atlantic (Fig. 1) over the past 16 million years (My), including the mid and late Miocene. To further evaluate proxy fidelity, we determined $\Delta_{47}$ temperatures of a monospecific *Coccolithus pelagicus* sediment trap in the Iceland Sea for which remotely sensed temperatures are well-constrained. We additionally estimated sea surface temperatures (SSTs) by applying widely-used calibrations[20–22] to U$^{k'}_{37}$ indexes measured on the same downcore ODP site 982 samples, as a comparison to our coccolith $\Delta_{47}$ calcification temperature record. Although alkenones and coccoliths are both produced by coccolithophores, we find significant differences in absolute temperatures amongst these proxies throughout the analyzed time interval. Our findings provide the first $\Delta_{47}$-based coccolith temperature reconstructions from the North Atlantic that align with Miocene climate model simulations suggesting only modest, rather than extreme high-latitude warmth, and underscore the need for continuous reassessment of both emerging and established SST proxies.

## Results and discussion
### Coccolith clumped isotope temperatures

The $\Delta_{47}$ temperature of the monospecific *C. pelagicus* sediment trap in the Iceland Sea (7.41 ± 4.4 °C; 95% confidence interval, CI; Fig. 2a), a sample produced by an unusual calcite bloom between June and July 1999, closely agrees with the AVHRR satellite-derived production temperatures of 6.74 °C[23]. This further supports the applicability of coccolith $\Delta_{47}$ as a reliable proxy of calcification temperatures. Our North Atlantic downcore $\Delta_{47}$ temperatures are from 91–98% pure and well preserved coccolith separations (2–10 μm; Fig. 3) and show mid-Miocene peak temperatures of 18.3 ± 5.0 °C, and a gradual cooling of ~9.0 °C from the Miocene to the quaternary (Fig. 2a). Absolute $\Delta_{47}$ temperatures are similar between the pure coccolith (2–10 μm) and the <11 μm size fractions. This implies that at this site and since the mid-

Miocene, neither foraminifera fragments (10–11 μm; Supplementary Fig. 1) nor potentially diagenetically-formed small unidentifiable carbonate <2 μm in size significantly affected the calculated temperatures.

### Coccolith $\Delta_{47}$ suggest a 10 °C colder North Atlantic compared to U$^{k'}_{37}$

When assessing the fidelity of coccolith $\Delta_{47}$ temperatures, several lines of evidence suggest that coccolith $\Delta_{47}$ reflect the primary calcification temperature of coccoliths with negligible influence from variable vital effects, diagenetic overprinting, nor cold biases. The samples of the fraction 2–10 μm not only consist of unprecedented highly pure (91–98%) coccoliths but are also dominated (78–93%) by the same *Reticulofenestra* taxa that produce alkenones (Supplementary Fig. 2). Given the dominance of these species, we expect that any cold bias in coccolith $\Delta_{47}$ temperatures due to assemblage variability remained well-below analytical detection. Despite the consistency in coccolith $\Delta_{47}$ temperature dependence across species[14,18], and the so far proven reliability of $\Delta_{47}$ calcification temperatures from mixed species[17,19], lateral advection of other coccoliths typical of more subpolar areas, like *C. pelagicus*, could introduce a cold bias. We evaluated this potential cold bias in $\Delta_{47}$ temperatures from the sample with the highest contribution of this species (10.6%) at ~14 Ma and show that this contamination would lead to a temperature underestimation of less than ~1 °C (Supplementary Note 1).

A cold temperature bias could in principle be introduced by the presence of diagenetic calcite formed at the seafloor. However, our data and analyses suggest only small contributions of authigenic carbonate in our coccoliths (Supplementary Note 2). Using estimates of the maximum amount of authigenic calcite in our samples (2.8–8.1%, Supplementary Table 1; Supplementary Figs. 3 and 4), the $\Delta_{47}$ diagenesis model of Stolper et al.[24] predicts cold biases of <2 °C (Supplementary Fig. 5), which we consider the uppermost limit on the potential influence of diagenesis on $\Delta_{47}$ temperatures. Moreover, the model indicates that diagenesis would lead to an increase in temperature offsets compared to the alkenone data with increasing sediment age, as older samples would undergo more extensive diagenetic alteration (Supplementary Fig. 5). However, this is not observed in our data, where $\Delta_{47}$ offsets relative to alkenone temperatures are consistent throughout the record. A negligible cold bias from authigenic carbonate in our record is further supported by the Sr/Ca ratios of our pure coccolith fractions (Supplementary Table 2), which are in the range of those typical of cultured coccoliths, sediment traps and sediment cores[25], and are 100-fold higher than those expected from abiogenic calcite[26]. Scanning Electron Microscopy (SEM) show evidence of some dissolution in all samples. To date, there is no evidence that partial dissolution would bias coccoliths $\Delta_{47}$. Coccoliths are chemically homogeneous crystals[27], generated within one hour[28], and are protected by polysaccharides[29,30]. This makes it unlikely that partial removal of calcite from etching in coccoliths would cause significant alteration of their $\Delta_{47}$ values, although further research might be needed to confirm this conclusion.

Coccolith $\Delta_{47}$ calcification temperatures over the last 16 My are on average ~9 °C colder than those derived from U$^{k'}_{37}$ from the same samples (Fig. 2a, b). The above discussion of potential cold bias sources in our coccolith $\Delta_{47}$ calcification record shows that, if present, they could only explain a small part of this ~9 °C difference (smaller than current $\Delta_{47}$ analytical errors). We propose, in agreement with coccolith core top $\Delta_{47}$ and sediment trap studies[17,19], that such differences can be at least in part explained by the calibration approaches applied to these proxies, in addition to non-thermal mechanisms potentially driving extra variability in alkenones.

We then assessed if alkenones records could overestimate North Atlantic temperatures. If alkenones were produced under the same conditions (i.e., season, depth, light, nutrients, and growth phase) during which coccolithophores calcify, we would expect similar

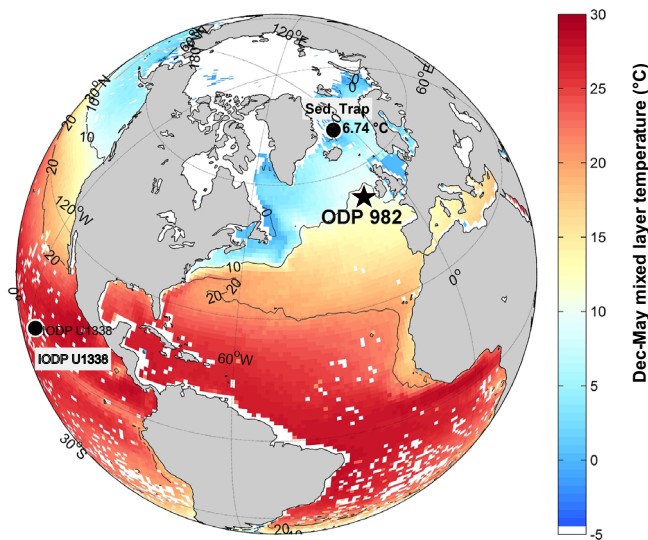

**Fig. 1 | December to May mixed layer depth temperature map with the location of sites discussed in this study.** Black star indicates ODP site 982, from which coccolith clumped isotope and alkenone temperatures were obtained for the last 16 My. Black dots indicate the location of the tropical IODP site U1338 and the sediment trap in the Iceland Sea (modern temperature 6.74 °C), the last from which coccolith clumped isotope temperatures were also measured. The figure was plotted using MATLAB based on data from monthly mixed layer depth temperature from the Global Ocean Surface Mixed Layer Statistical Monthly Climatology[68].

absolute temperature estimates from both proxies. However, despite sharing similar trends and being correlated (Fig. 2 and Supplementary Fig. 6), coccolith $\Delta_{47}$ calcification temperatures are significantly colder than alkenone temperatures estimated using calibrations based on regressions of $U_{37}^{k'}$ against SSTs[20,21]. From existing alkenone culture calibrations (analog to $\Delta_{47}$ calibrations), only the batch *Emiliania huxleyi* (strain 55a) calibration[22] agrees with calibrations based on SSTs[20,21]. Several other culture studies on different strains of *E. huxleyi* and *Gephyrocapsa oceanica* show different alkenone unsaturation calibrations to growth temperatures[31]. When applied to our $U_{37}^{k'}$ dataset, there are up to 8.0 °C differences among these culture experiments and surprisingly, all yield even warmer temperatures than the modern SST and the published coccolith $\Delta_{47}$ core top temperature[17] (Fig. 2a and Supplementary Table 5). Similarly, the application of the alkenone calibration based on field-measured temperatures[32] to $U_{37}^{k'}$ of the Holocene sample[17] result in even larger (>7 °C) overestimates of modern SSTs (Fig. 2a). These large differences in the sensitivity of $U_{37}^{k'}$ to cultured temperature highlight the importance of understanding potential nonthermal effects in alkenones, such as nutrient stress and the related cellular growth phase (exponential, stationary), light availability, type of culture method (batch or continuous), or physiological aspects unique to a species or strain[33–35]. In addition to nonthermal effects, $U_{37}^{k'}$ measured on fossil samples can be affected by preferential degradation of $C_{37:3}$ in highly oxygenated waters[36–38], which could be particularly important for the North Atlantic. Moreover, temperature biases could also arise from laterally-transported alkenones attached to coarse easily-transported sediment fractions[39], potentially produced in warmer areas. Both mechanisms could introduce a warm bias in our alkenone-derived temperatures. Improving our knowledge on aspects like the utility of synthesizing alkenones, cellular production pathways, and all possible non-thermal mechanisms would help clarifying absolute temperatures (Fig. 2a, b) and which calibrations are most appropriate for a given oceanographic setting.

Widely-used alkenone calibrations used for temperature reconstructions in the North Atlantic are based on annual (core top[20]) and warm season (August–October; BAYSPLINE[21]) temperatures at 0 m. In contrast, $\Delta_{47}$ calibrations are based on measured or inferred temperatures during carbonate formation. Consequently, in places or time intervals in which coccolith biomineralization (and alkenone formation) occurs at depth and/or during cooler seasons, the application of coretop[20] and BAYSPLINE[21] calibrations to $U_{37}^{k'}$ records is expected to overestimate mean annual SSTs and produce warmer estimates than actual calcification temperatures derived from $\Delta_{47}$. We suggest that since the mid-Miocene, absolute calcification temperatures at ODP site 982 are better represented by coccolith $\Delta_{47}$, and that the application of the widely used calibrations based on regressions of mean annual or warm season SSTs to alkenones[20,21] likely leads to overestimated temperatures for the season and depth of production.

Coccolithophore production in the modern North Atlantic is the highest between the colder winter and spring seasons[40], and there is no evidence for peak production during the warmer August–October period[40–45]. Moreover, coccolith $\Delta_{47}$ was suggested to represent mixed layer, rather than surficial temperatures for the same site[17]. While the maximum temperature effect of production deeper than the sea surface is relatively small for this location, due to its weak thermocline (maximum of -1.6 °C, assuming deepest production at 100 m; Supplementary Note 3 and Supplementary Table 3), the maximum temperature effect of applying alkenone calibrations based on SSTs outside the actual season of production is larger (up to 3 °C; Supplementary Note 4 and Supplementary Table 4). For the mid-Holocene ODP site 982, published alkenone-derived temperatures using the core-top[20] and BAYSPLINE[21] calibrations were up to 5.9 °C warmer than both modern SSTs during the season of production and coccolith $\Delta_{47}$ calcification temperatures[17]. Assuming the modern North Atlantic represents well the mid-Holocene conditions at the same site, the maximum effects of applying alkenone calibrations based on (1) SSTs rather than on temperatures at depth of production and (2) SSTs during a warmer season than that of production, can explain up to 78% of the difference in published mid-Holocene absolute temperature estimates between coccolith $\Delta_{47}$ and alkenones[17]. An increasingly stratified North Atlantic during warmer past intervals, or a larger temperature difference between seasons, could exacerbate the depth and the season of production effects, therefore increasing differences of estimated temperatures between proxies.

An alternative empirical alkenone calibration based on season of production temperatures at depth, rather than on annual or August–October SSTs, was recently proposed[17]. This calibration employs a subset of sites from the broader global alkenone calibration set[21], for which the season and depth of production can be inferred to be similar to those at geographically proximal core top sites for which coccolith $\Delta_{47}$ were determined[17]. This empirical calibration regresses these $U_{37}^{k'}$ values to the temperatures at the depth and season of production inferred from the core top coccolith $\Delta_{47}$ dataset[17]. Applying this calibration to our ODP 982 $U_{37}^{k'}$ values and those previously published for our site[1], we obtain absolute alkenone-derived growth temperatures that agree much better with the absolute values of our coccolith $\Delta_{47}$ record. The same is true when this calibration is applied to the higher resolution ODP site 982 $U_{37}^{k'}$ Miocene values of the study of Super et al.[8], which decreases average alkenone temperatures by ~6.6 °C (Fig. 4). Although we recognize that a much larger dataset would be required to make such a calibration more robust and widely applicable for reconstructions, these results suggests that when depth and season of production of coccolithophorids are considered in the calibrations, a large part of the observed discrepancies in absolute values between coccolith $\Delta_{47}$ and alkenone proxies are resolved. This adds confidence to our conclusion from coccolith $\Delta_{47}$ that the euphotic North Atlantic was likely ~9 °C colder than what alkenone temperatures suggest when applying conventional calibrations[20,21]. ODP site 982 coccolith $\Delta_{47}$ temperatures and recalibrated alkenone temperatures are also more compatible with the deep-sea benthic

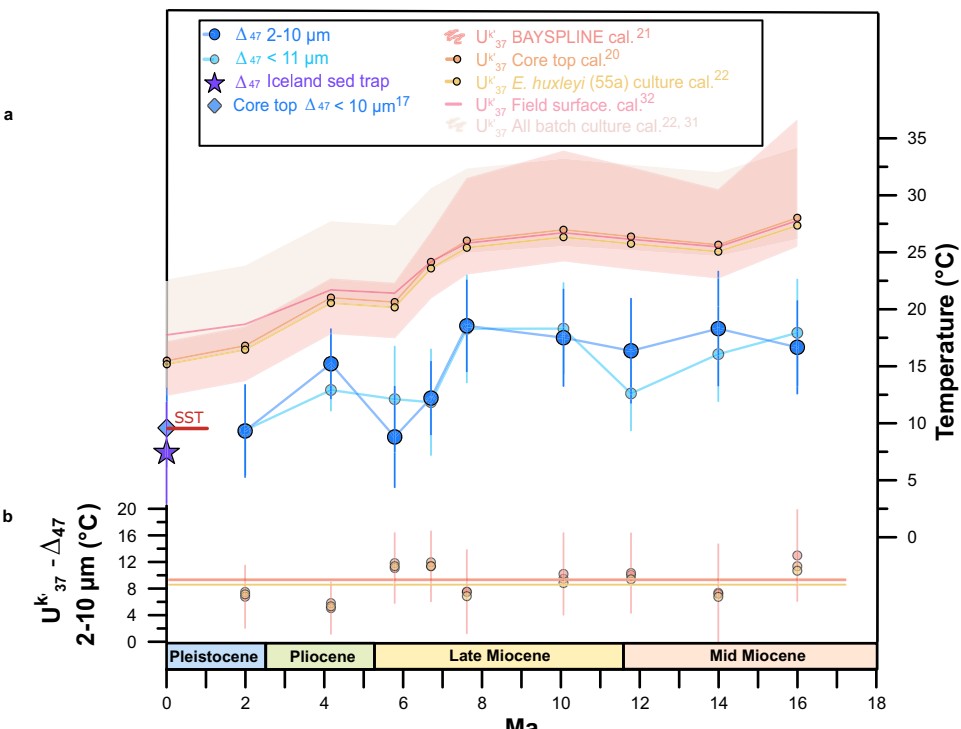

**Fig. 2 | Coccolith clumped isotope and alkenone temperature evolution in the North Atlantic (ODP site 982) over the last 16 My. a** $\Delta_{47}$ calcification temperatures (cold colors) from the pure coccolith 2–10 (blue dots) and the <11 μm size fractions (light blue dots), and alkenone-derived temperatures (warm colors) from the same samples calculated using the core top[20] (orange dots), BAYSPLINE[21] (pale pink shade is the 95% confidence interval (CI)), field surface water[32], *E. huxleyi* 55a batch culture[22] (pale yellow dots), and ten further culture calibrations[31] (max. and min. values within the pale pink shaded area). Alkenone temperatures also calculated from the published coretop $U^{k'}_{37}$ value[17] of our same site. Sediment trap $\Delta_{47}$ temperatures from the Iceland Sea *C. pelagicus* sample shown as a purple star. Coccolith $\Delta_{47}$ calcification temperatures from a core top (<10 μm) at our same location[17] (blue diamond) fit well modern ocean sea surface temperatures (SST red horizontal line). Error bars in coccolith $\Delta_{47}$ calcification temperatures record denote the 95% CI. **b** Average temperature differences between our alkenone-derived records calculated using the core top[20] (orange dots), BAYSPLINE[21] (pink dots) and *E. huxleyi* 55a batch culture[22] (pale yellow dots) calibrations, and the coccolith $\Delta_{47}$ record from the pure 2–10 μm size fraction. Horizontal dashed lines denote average temperature differences calculated using results from all samples. Error bars for differences between alkenone BAYSPLINE and coccolith $\Delta_{47}$ temperatures are calculated propagating errors from both records.

foraminifera $\Delta_{47}$ temperature reconstructions from downstream North Atlantic sites likely proximal to deep-water formation areas[11,12]. For the Miocene, most available temperature records are alkenone-based[4]. If similar findings of cooler coccolith $\Delta_{47}$ production temperatures compared to alkenones were reproduced at other high latitude sites during the Miocene, they would have important implications in our understanding of future high latitude amplification.

Miocene TEX$_{86}$ absolute temperatures for ODP site 982 calculated using a shallow subsurface calibration[46] show generally similar (or slightly colder) values compared to a location 14.7° further south in the subtropical gyre[8], and generally fall between our alkenone and coccolith $\Delta_{47}$-derived temperature estimates (Supplementary Fig. 7). Similar temperatures at ODP site 982 and at the subtropical gyre would be possible under extreme high latitude amplification. Alternatively, it is also possible that there was still a latitudinal thermal gradient between these sites that cannot be discerned by TEX$_{86}$ at these locations or time intervals, potentially due to similar challenges in the attribution of the production depth and season. If accurate, these TEX$_{86}$ "warm subsurface" values would indicate that both alkenone and coccolith $\Delta_{47}$ temperatures underestimate euphotic zone temperatures. However, when we apply the subsurface TEX$_{86}$ calibration[47] to this dataset [8], absolute values mostly fall under those suggested by coccolith $\Delta_{47}$ temperatures (Supplementary Fig. 7). Considering that the subsurface calibration[47] has its highest occurrence at integrated 0–550 m water depths, and that the difference between SSTs and mean 0–550 m temperatures in the modern ocean during the winter-spring season is very small (-0.36 °C) due to the small thermocline, slightly

lower but similar TEX$_{86}$ estimates compared to the euphotic zone (-71 m) coccolith $\Delta_{47}$ temperatures are expected (Supplementary Fig. 7). While this discussion provides some context for TEX$_{86}$ temperature comparisons to our proxies, we highlight that alkenone and coccolith $\Delta_{47}$ temperatures can be readily compared because they derive from the same organism. Detailed comparison of absolute coccolith $\Delta_{47}$ temperatures with other records from proxies based on other organisms, like TEX$_{86}$, would require a thorough analysis not only of calibrations but also poorly constrained differences in the ecology of the biomarker producing archaea, which is beyond the scope of this paper.

### Modest, not extreme northern high latitude amplification over the mid-late Miocene: perspectives for model-coccolith $\Delta_{47}$ data comparisons

Our coccolith $\Delta_{47}$ record provides estimates of North Atlantic absolute temperatures from the mixed layer winter-spring season since the mid-Miocene and provides a new target for paleoclimate model-data comparisons. For locations with a strong seasonal temperature cycle, robust model-data comparisons require a clear attribution of season, but also of depth of the signal. Our analysis suggests these criteria are well met for coccolith $\Delta_{47}$ calcification temperatures in the North Atlantic, because they are coherent with the known production regime. Our $\Delta_{47}$ temperature record is the first to agree (for late Miocene), and the closest to match (for middle Miocene) modeling studies (Fig. 5 and Supplementary Fig. 9), including that of the first attempt of a Miocene multi-model comparison[4,6,48].

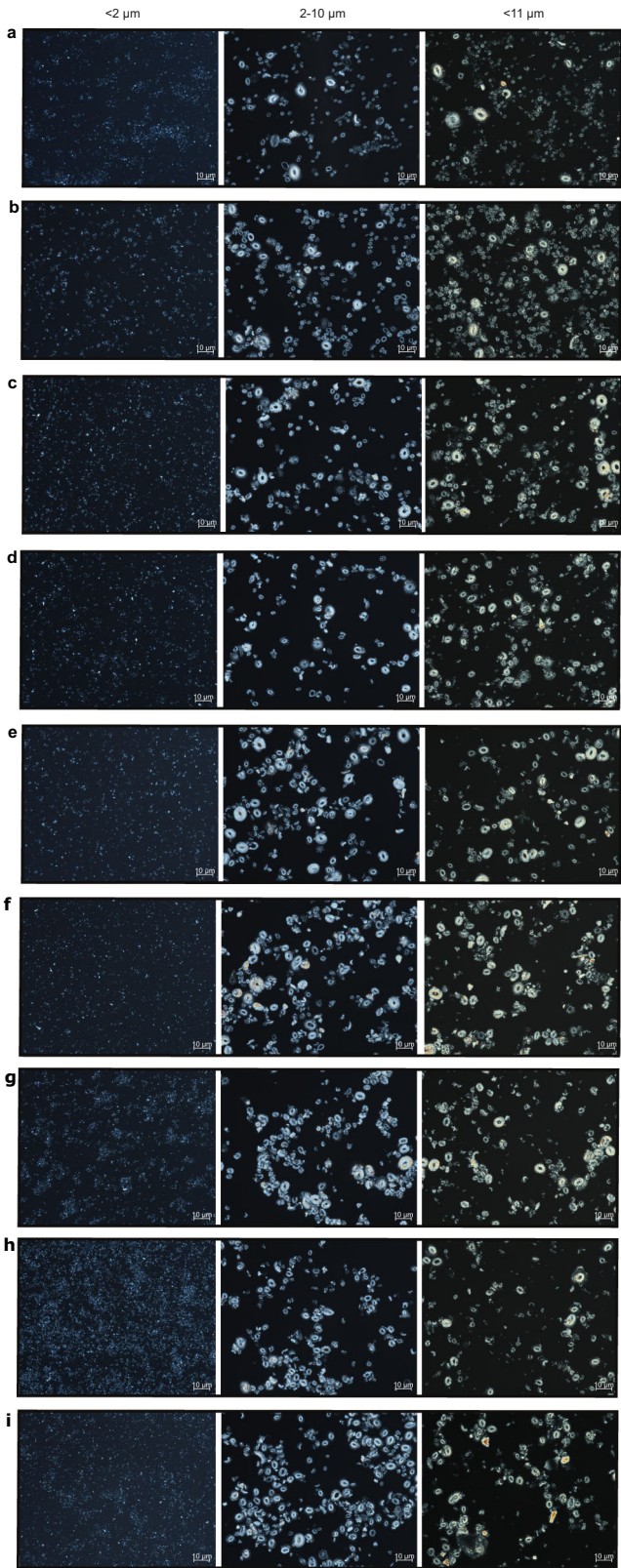

**Fig. 3 | Light microscope images of <2 μm, pure coccolith (2–10 μm), and <11 μm size fractions of sediment samples from ODP site 982. a** 1.99 Ma. **b** 4.17 Ma. **c** 5.79 Ma. **d** 6.71 Ma. **e** 7.61 Ma. **f** 10.07 Ma. **g** 11.78 Ma. **h** 13.99 Ma. **i** 16 Ma.

The flattening of the Miocene latitudinal thermal gradient and extreme high latitude amplification in the Atlantic[1,7,8], especially for the warmest mid-Miocene, has been a major paleoclimate conundrum because climate models struggle to achieve such warm high latitude

temperatures (e.g., refs. [4],[6],[48],[49]), suggesting complications with proxy interpretations or missing physics in climate models. The study that concluded a persistent high latitude amplification in the Pacific since the late Miocene based their calculations in comparisons of their multiproxy Western Pacific Warm Pool stack with North Pacific alkenone-derived SSTs[50]. The latter absolute values have only been successfully simulated by the COSMOS model, that also significantly overestimates tropical temperatures[4]. Since there are no core top coccolith $\Delta_{47}$ data for the North Pacific[17], it is currently not possible to estimate if and by how much season and depth of production may be biasing alkenone absolute temperatures in this ocean setting. Our data and analyses are specific to the North Atlantic and, therefore, our conclusions currently cannot conclusively be extended to other oceanic regions. Nevertheless, the significant warm biases (up to ~6 °C) in alkenone-derived SSTs from surface sediments in the North Pacific and the high sensitivity of these biases to seasonality changes[51], suggest that alkenone-based reconstructions in this area could also be biased and therefore not appropriate to calculate amplification. The same argument is valid for the latest North Pacific alkenone-derived SST record that includes the Late and mid-Miocene[52]. Our comparatively smaller coccolith $\Delta_{47}$ dataset hampers the calculation of reliable high latitude amplification from similar comparisons to tropical temperatures as conducted by Liu et al.[50]. However, increasing the resolution of coccolith $\Delta_{47}$ in high latitudes would allow them in the future.

Miocene extreme polar amplification has been best simulated using $CO_2$ concentrations around the maximum values (or higher) than those suggested by proxies[4,48,53], but high latitude warmth in places like the North Atlantic (and the high latitude southern hemisphere) continues to fall short in model simulations, while tropical temperatures tend to be overestimated. Our North Atlantic coccolith $\Delta_{47}$ temperatures suggest there is an overestimated Miocene high latitude warmth associated to the alkenone proxy interpretation, but extreme amplification could additionally be caused by underestimates in tropical temperatures, mostly derived from alkenones as well[4]. A cold bias in tropical and subtropical regions could arise from the lower $U^{k'}_{37}$ sensitivity to high temperatures[34], and the analytical problem to detect $C_{37:3}$ alkenones when $U^{k'}_{37}$ approaches the limit of one[54]. In line with this, a less extreme polar amplification for the Pacific Ocean was reported for the late Miocene when instead of alkenones, Mg/Ca from a mixed layer foraminifera was used as proxy to reconstruct tropical temperatures[53]. On the other hand, it is also possible that the application of traditional calibrations based on SSTs[20,21] to alkenone data from tropical, stratified oceans lead to larger overestimates of SSTs compared to more mixed areas like the North Atlantic[17]. If this was also valid for the Miocene, it would result in increased polar amplification. Regardless of the absolute values of tropical temperatures, comparisons to our North Atlantic coccolith $\Delta_{47}$ temperature record produce a more modest polar amplification than when compared to alkenone-derived temperatures.

The best fit with current proxy data for the late Miocene was achieved with the NorESM-L model set at 560 ppm of $CO_2$ (higher than proxy estimates), which includes an improved representation of cloud microphysics and led to the best capture of polar amplification during the Eocene[55]. Mean annual temperature late Miocene simulations from this model show latitudinal temperature gradients between ODP site 982 and the Eastern Equatorial Pacific (EEP) of -14.7 °C, slightly smaller than those of the modern ocean (-15.4 °C). Ideally, model-data comparisons should be based on the same season, and consider the depth from which the proxy temperature signal originates. However, the majority of modeled seasonal and water depth temperatures for the Miocene are not publicly available and comparisons have been traditionally made with proxy data without considering potential proxy seasonality or depth biases (e.g., North Atlantic alkenone-derived SST using the BAYSPLINE calibration indicate August–October SSTs). Unfortunately, neither the winter-spring or the water column

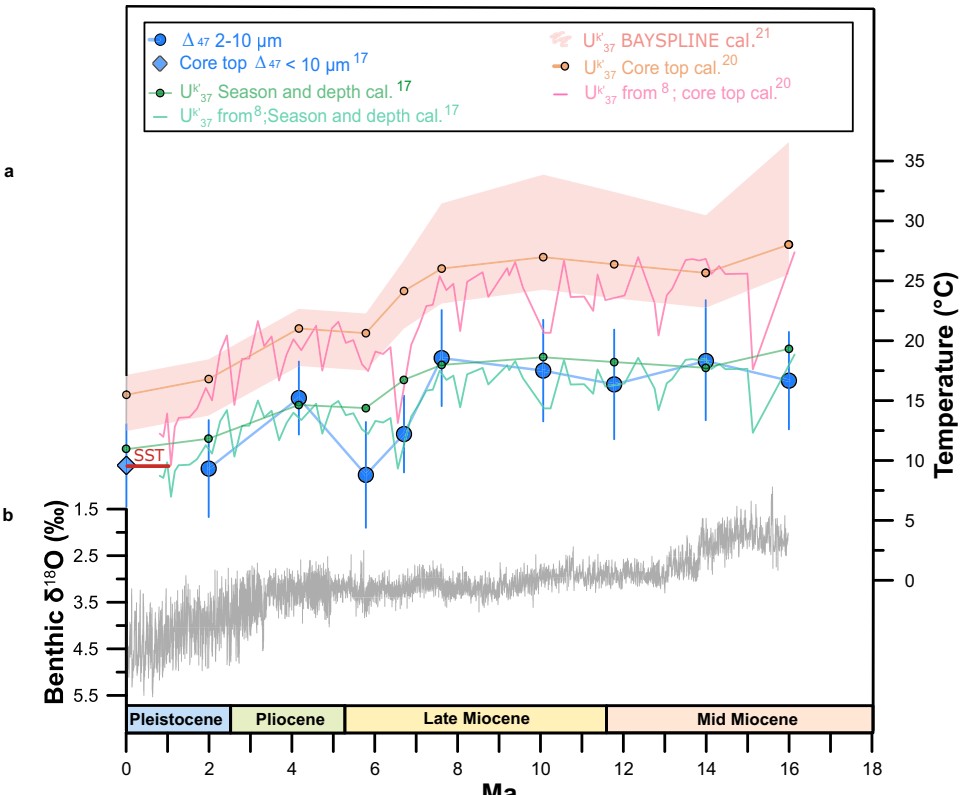

**Fig. 4 | ODP site 982 absolute coccolith clumped isotope calcification temperatures agree better with alkenone temperatures calculated using a calibration that considers season and depth of production. a** $\Delta_{47}$ calcification temperatures (2–10 µm; blue dots), and alkenone-derived temperatures from this study as in Fig. 2, including estimates using the core top[20] (orange dots), BAYSPLINE[21] (pale pink shade), and a calibration that considers the season and depth of production ([17] green dots). From the study of Super et al.[8]: alkenone temperature record applying the core top[20] (pink line) and a calibration that considers the season and depth of production[17] (blue-green line). The latter generally agrees with absolute temperatures derived from coccolith $\Delta_{47}$ (blue dots) and those derived from our alkenones calculated using the same season-depth calibration (green dots). Pale pink shaded area represents the 95% CI according to the BAYSPLINE calibration. Alkenone temperatures also calculated from the published coretop $U^k_{37}$ value[17] of our same site. Coccolith $\Delta_{47}$ calcification temperatures from a core top (<10 µm) at our same location[17] (blue diamond) fit well modern ocean sea surface temperatures (SST red horizontal line). Error bars in coccolith $\Delta_{47}$ calcification temperatures record denote the 95% CI. **b** Cenozoic Global Reference deep-sea benthic foraminifer oxygen Isotope Dataset (CENOGRID)[3].

temperature structure from this model are publicly available. Therefore, we take the traditional approach and directly compare model mean annual temperatures with coccolith $\Delta_{47}$-derived winter-spring temperatures (productive season). In addition, we have attempted to back-calculate surface mean annual temperatures from our coccolith $\Delta_{47}$ to compare to modeled mean annual temperatures, assuming the difference between surface mean annual temperatures and winter-spring temperatures at production depth since the Miocene was similar to that of the modern ocean (seasonal effect = -0.97 °C; depth effect = -0.34 °C; total = 1.31 °C; Supplementary Note 5). While this assumption could be invalid in a changing ocean since the Miocene, given the different climate state and continental distribution, leading to likely varying temperature contrast between seasons, changes in coccolithophore season and depth of production, and changes in the relationship between SSTs and temperatures at depth at our site, it is the only way we can currently directly compare measured coccolith $\Delta_{47}$ temperatures and modeled mean annual temperatures. Assuming that alkenones represent well temperatures of the EEP[56], latitudinal temperature gradients for the late Miocene calculated using our North Atlantic coccolith $\Delta_{47}$ (-13.5–12.1 °C) are much more consistent with climate models than those derived from the average of all other proxies available for our location (-4.7 °C including our alkenone record) (Fig. 5 and Supplementary Fig. 9). Although the traditional interpretation of alkenone calibrations to SSTs and non-thermal processes could lead to overestimates of EEP temperatures as well[17], comparisons to high latitude temperatures does not change our

conclusions of a better match of coccolith $\Delta_{47}$ calcification temperatures with models in the North Atlantic high latitudes.

For the mid-Miocene, even $CO_2$ concentrations of 850 ppm were not able to reproduce the even flatter temperature gradient shown by proxy data (best fitting models: HadCM3L[4]), and EEP temperatures were overestimated by >3 °C, which could be related to model limitations in simulating upwelling regions. Lower $CO_2$ concentrations improve tropical temperature simulations, but in these simulations the North Atlantic is even colder than the modern ocean. The multiple optimization Earth System model of mid-Miocene based on cGENIE required $CO_2$ concentrations as high as 1120 ppm to achieve the highest North Atlantic temperatures without overestimating tropical temperatures, simulating a latitudinal thermal gradient of 15.8 °C[48]. In the case of the mid-Miocene, there is one recent water isotope enabled Community Earth System Model output (MIO400 CI; 400 ppm of $CO_2$) that includes open access to mean annual water column temperatures[6]. Surficial mean annual temperatures from this model (16.2 °C) only differ by 0.9 °C from the mean annual temperatures of the upper 0–70 m water column (modern euphotic zone limit for site 982) (15.3 °C), matching our coccolith $\Delta_{47}$ calcification temperatures for the mid-Miocene. While the gradients between EEP alkenone-derived temperatures and our North Atlantic coccolith $\Delta_{47}$ winter-spring and mean annual temperature records are more similar to models[4,6,48] (-11.2 °C and 9.9 °C, respectively), the gradient suggested by other published proxies is negligible (2.7 °C) (Fig. 5, Supplementary Fig. 9).

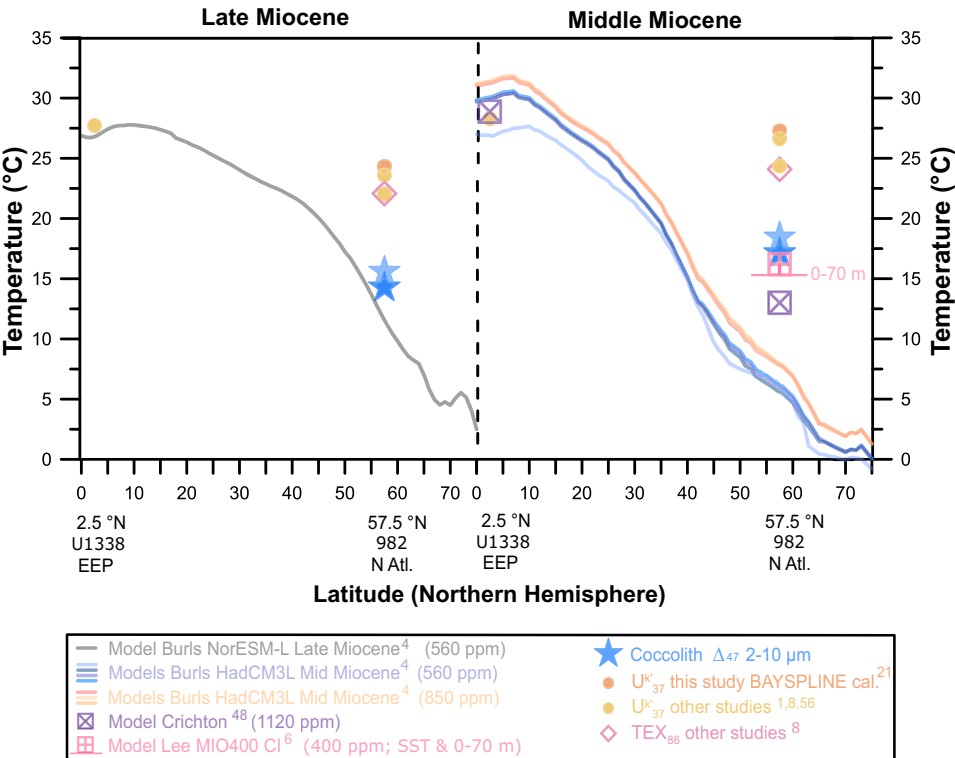

**Fig. 5 | Late and mid-Miocene latitudinal thermal gradient shown by coccolith clumped isotopes, alkenones and model simulations.** While alkenone temperatures calculated using widely-used calibrations suggest a small (late Miocene: from 11.6 to 5.33 Ma) or negligible (mid-Miocene: from last sample at 16–11.6 Ma) latitudinal thermal gradient ([1,8,56], and this study), coccolith clumped isotopes (as winter-spring at depth of production-dark blue star- and approximated surficial mean annual temperatures -pale blue star- temperatures) and Miocene model simulations[4,6,48] suggest a more modest polar amplification and a smaller flattening of the latitudinal thermal gradient. Modeled mean annual average 0–70 m water column temperatures from Lee et al.[6] shown as a horizontal pink line below mean annual sea surface temperatures for ODP site 982, from which it differs by 0.9 °C. 95% confidence intervals for clumped isotope measurements are between 4 and 4.5 °C and for alkenone temperatures (BAYSPLINE calibration)[21] are between 2.8 and 5.9 °C. Refer to Supplementary Fig. 9 to see the representation of this figure, including errors (95% CI) for the data produced in this study.

This analysis shows that coccolith $\Delta_{47}$ calcification temperatures are more consistent with modeled late and mid Miocene latitude thermal gradients and suggest a lower degree of high latitude amplification for the North Atlantic than that inferred from other proxy data.

Simulating the extreme high latitude warmth suggested by widely-used temperature proxies during past warm intervals, especially during the Miocene, has proven to be very challenging for the climate modeling community. The debate over high latitude amplification exists for other time intervals like the Eocene, although new modeling attempts have been able to better reproduce high latitude warmth[55,57]. During the Eocene, however, it was paleoceanographic forcing, rather than $CO_2$ alone, that may have contributed more to high latitude warmth compared to the Miocene[4]. Our downcore record of pure coccolith $\Delta_{47}$ calcification temperatures in the North Atlantic over the mid to late Miocene is the first to show absolute values that agree much better with model simulations, suggesting that the North Atlantic was ~9 °C colder than what other proxies previously showed. We suggest that it is worth exploring coccolith $\Delta_{47}$ as a proxy to test high latitude amplification in other regions and also further back in time. If the more modest mid and late Miocene high latitude warmth shown by our coccolith $\Delta_{47}$ is observed in other high latitude sites, the conclusion of modest, not extreme amplification, would provide a more optimistic perspective of high latitude climate response to anthropogenic $CO_2$ emissions in the future than implied by proxy data in the past, while underscoring the necessity to better understand the mechanisms affecting all existing temperature proxies at different locations and times.

## Methods

### Oceanographic setting in the North Atlantic

ODP site 982 is located in the North Atlantic (Rockall Plateau, 57° 31.002' N and 15° 51.993' W; water depth 1134 m; Fig. 1). Its paleogeographical location has not changed significantly in the last 15 My[1] and sediment is carbonate rich (86%), making it ideal for achieving pure coccolith samples. We used nine samples with depths ranging between 43.99 and 524.55 m (mcd), corresponding to ages between 1.99 and 16 Ma. The age model until 5 Ma was based on the correlation of benthic foraminiferal $\delta^{18}O$ from ODP site 982 and those of the LR04 stack, while after 8 Ma, it was based in biostratigraphy[1]. Between 5 and 8 Ma, we used the latest age model derived from high resolution XRF core scanning data and benthic foraminifera $\delta^{18}O$ and $\delta^{13}C$ astrochronology[58]. We also used a sediment trap sample from the Iceland Sea (70.23° N; 9.75° W; 1884 m)[23], which in July 1999 registered the largest surface bloom ever recorded in this area, containing 99% of the subpolar north Atlantic *C. pelagicus*.

### Alkenone thermometry

Bulk sediments were freeze-dried, and the total lipid extract was obtained via accelerated solvent extraction following methods detailed in Mejia et al.[17]. After saponification using a 0.5 M solution of KOH in MeOH: $H_2O$ (95:5), the neutral alkenone-containing fraction was extracted with toluene and further purified via silica gel column chromatography. The ketone fraction containing alkenones was measured at ETH Zürich using a Thermo Scientific Trace 1310 gas chromatograph coupled to a flame ionization detector, as shown in

Guitián et al.[59]. The $U_{37}^{k'}$ ratio was calculated from the abundance of $C_{37:2}$ and $C_{37:3}$, from which SSTs were derived using the core top[20], the 55a *Emiliania huxleyi* batch culture[22] and the BAYSPLINE[21] calibrations. In-house alkenone standard repeated measurements yielded a precision of 0.012 $U_{37}^{k'}$ units (0.36 °C calculated with the core top[20] calibration).

## Coccolith clumped isotope thermometry

**Sample processing.** After lipid extraction for alkenone analyses, samples were microfiltered in ammonia solution (0.5%) at 11 μm to obtain a coccolith-enriched fraction. To avoid potential interference from organics during $\Delta_{47}$ analyses, we eliminated the remaining organic matter using buffered 10% $H_2O_2$, as described in Mejia et al.[17]. No effects in coccolith stable or clumped isotopes were reported using this method[17]. Since diagenetic processes are more prone to happen in the smallest fragments, whose source is also impossible to identify, we used centrifugation techniques (seven repetitions at 2300–2800 RPM for 2 min) to remove the <2 μm size fraction[60]. We then produced a pure coccolith fraction (2–10 μm) by extensive microfiltration at 2 and 10 μm. The remaining 10–11 μm size fraction showed enrichment in fragments of foraminifera (Supplementary Fig. 1). Coccolith purity of the 2–10 μm fractions and species' assemblages were determined using light microscopy. To test whether small calcite of unidentifiable origin (<2 μm) and large calcite (10–11 μm) have a significant effect in coccolith $\Delta_{47}$ temperatures, a small aliquot from the extracted bulk sediment was sieved at 11 μm using ethanol, and then oxidized with $H_2O_2$ as described above. Before $\Delta_{47}$ analysis, all samples were rinsed with Mili-Q, dried at 50 °C and homogenized.

**Evaluation of diagenetically-sourced cold bias.** The presence of abiogenic calcite produced at depth and at colder temperatures compared to the original coccolith signal can introduce a cold bias in $\Delta_{47}$ temperatures. The degree of secondary overgrowth on coccoliths was evaluated both by trace element analyses and by SEM. We used weak acetic acid (0.4 M) to dissolve 50–100 μg of the pure coccolith (2–10 μm) and the <2 μm size fractions. Sr, Mg and Al/Ca ratios were determined using an Agilent 8800 Triple Quadrupole ICP-MS at ETH Zürich, following the intensity ratio calibration described in ref. 25.

To date, there are no techniques able to quantify authigenic calcite in coccoliths. Most studies provide only qualitative descriptions of calcite preservation. Instead, here we produced a very conservative estimate of the maximum diagenesis effect by applying the geometrically-calculated coccolith volume plots of Young and Ziveri[61] to calculate the % volume affected from the % of authigenic overgrown area obtained by SEM. Then, this was extrapolated from the center to the edge of the coccolith observed in cross-section, assuming a maximum of half of the calcite was affected by diagenesis (Supplementary Table 1). We applied this method to 13–26 single coccoliths for each sample (average of 18 coccoliths per sample). We also determined the maximum potential effect of the degree of diagenesis on our $\Delta_{47}$ temperature estimates by implementing the $\Delta_{47}$ diagenesis model of Stolper et al.[24] (Supplementary Fig. 5), which uses the same model construction as the diagenesis models of refs. 62,63 (details in Supplementary Note 2), and applied it to our ODP site 982 coccoliths.

**Clumped isotope measurements and temperature estimates.** Measurements were conducted using a Kiel IV-Thermo Scientific MAT 253 at ETH Zürich, following the LIDI protocol[64], and the procedures described in Mejia et al.[17], including a Porapak Q trap (−40 °C) to eliminate residual halo/hydrocarbon and reduced sulfur compounds. Nine to 21 replicates of ~110 μg carbonate were measured depending on sample availability. Measurements were conducted over a period of 18 months, using the carbonate standardization scheme based on ETH-1 ($\Delta_{47}$ = 0.2052‰), ETH-2 ($\Delta_{47}$ = 0.2085‰), and ETH-3 ($\Delta_{47}$ = 0.6132‰)

standards[65]. Long term external reproducibility was monitored using the standard IAEA C2 (standard deviation: $\delta^{13}C$ = 0.02‰, $\delta^{18}O$ = 0.03‰; $\Delta_{47}$ = 0.03‰). Data processing was carried out with the software Easotope[66]. Measurements with $\Delta_{48}$ offset > 2 and 49 parameter values > 2‰ were eliminated because it was considered that they were affected by contamination[66].

As demonstrated by the core top coccolith $\Delta_{47}$ study[17], the application of abiogenic $\Delta_{47}$ calibrations to coccolith samples should be avoided, as they derive too cold temperatures that are found at water depths at which coccolithophores would not be able to photosynthesize. Similarly, the recent coccolith culture and sediment trap studies[18,19] show that coccoliths have a systematic offset from the generalized calibration that includes abiogenic samples[67], and that there is a consistent relationship between growth temperature and $\Delta_{47}$ across different species[14,18]. Here we use this coccolith culture $\Delta_{47}$ calibration[18] to calculate calcification temperatures from $\Delta_{47}$ of ODP site 982 coccoliths. The application of this coccolith calibration leads to absolute temperature values and trends that are remarkably similar to those obtained using the foraminifera calibration of ref. 11 (Supplementary Fig. 8). Further data on both cultured coccoliths and foraminifera would clarify if the magnitude of the offset to abiogenic carbonates may be shared by these biogenic carbonates.

## Data availability
All data generated are available as a supplementary file in this publication, and are deposited in Pangaea.de[69] in two separate datasets[70,71].

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

## Acknowledgements

This project has received funding from the European Union's Horizon 2020 research and innovation programme under the Marie Sklodowska-Curie grant agreement 795053, from ETH Zurich Core funding, and from MARUM through DFG Germany's Excellence Strategy, Cluster of Excellence "The Ocean Floor-Earth's Uncharted Interface" (EXC-2077, Project 390741603). A.F. acknowledges support from project PID2023-151870OA-I00 funded by MICIU/AEI/10.13039/501100011033 and by ERDF/EU. We thank laboratory technician Stewart Bishop, laboratory assistants Manuel Walde and Sarah Rowan, Steve Maganini, Dorinda Ostermann and Susumu Honjo for providing the sediment trap sample, and Iván Hernández-Almeida for assistance in depth-integrated variable calculations.

## Author contributions

L.M.M. and H.Z. developed the separation method; A.F. applied the diagenesis model; L.M.M. separated and cleaned the coccoliths, estimated authigenic carbonate; L.M.M. and M.J. measured clumped isotopes under the direction of S.B.; L.M.M. and M.J. prepared and measured samples for trace element analysis. L.M.M. and A.P.H. took the SEM pictures; H.Z. evaluated coccolith assemblages. L.M.M. purified alkenones, and L.M.M. and J.G. measured alkenones. L.M.M. wrote the paper with contributions from A.F., S.B., H.S., H.Z., and V.T.

## Funding

## Competing interests

The authors declare no competing interests.
