## [Transparent Peer Review File · Nature Communications]

Coccolith clumped isotopes reveal modest rather than extreme northern high latitude amplification during the Miocene

Corresponding Author: Dr Luz Maria Mejia Ramirez

Version 0:

Reviewer comments:

Reviewer #1

(Remarks to the Author)

In this study, the authors generate a North Atlantic temperature record spanning the past 16 million years using coccolith $\Delta 47$. This approach is relatively novel, and $\Delta 47$ is theoretically a proxy that is not affected by issues plaguing other proxies, making it a superior choice. The coccolith $\Delta 47$ temperature record is approximately 9°C lower than that of alkenone-based UK'37. Since both coccoliths and alkenones are produced by coccolithophores, these proxies should yield similar temperature estimates. The authors suggest that the discrepancy between the proxies is due to issues with UK'37 calibrations. Additionally, the coccolith $\Delta 47$ proxy outperforms commonly applied UK'37 and TEX86 in terms of agreement with climate models. $\Delta 47$ temperatures support a modest decrease in the latitudinal temperature gradient during the Miocene, in contrast to the extreme gradient flattening suggested by UK'37 and TEX86.

Despite numerous typos, the manuscript is well-written and enjoyable to read. This is one of the first reconstruction studies to use coccolith $\Delta 47$, scoring high on the novelty scale. Compared to other commonly used temperature proxies, $\Delta 47$ is not yet known to be affected by non-thermal issues that undermine the reliability of other proxies. As is common in proxy development, more issues may be uncovered later in its development stage. However, at this moment, coccolith $\Delta 47$ is a superior proxy, lending credibility to the resulting reconstructions. If the conclusion that UK'37 temperatures are severely overestimated proves to be true, this would have broad implications for paleoceanography across different timescales, given the widespread use of UK'37 in the community.

While I find the study timely and the results important, I am not yet fully convinced by some of the arguments presented. Some reasoning appears incomplete and requires further evidence and clarification. I hope the line-by-line comments will be helpful in strengthening these arguments when the authors revise the manuscript. The methods section is clear and sufficiently detailed for the work to be reproduced, but proxy errors should be considered in the model-data comparison.

The main finding of the study is that $\Delta 47$ temperatures are substantially lower than UK'37 temperatures. The authors did a great job at carefully considering and then ruling out potential cold bias in coccolith $\Delta 47$ data. By contrast, the discussion of UK'37 data is less in depth, drawing heavily from Mejia et al. (2023), which proposes that the application of UK'37 SST calibrations leads to temperature overestimation, as the proxy signal originates from subsurface waters in the tropics. Mejia et al. (2023) then generated a UK'37 production temperature calibration using subsurface temperatures (150m) in the tropics and SST in high latitudes, which naturally differs in sensitivity compared to those widely applied SST calibrations (i.e., SST latitudinal gradient > gradient between low latitude subsurface and high latitude SST). The authors then applied Mejia et al. (2023) UK'37 production temperature calibration to their Miocene record, which improves the match with the $\Delta 47$ record. However, this is to be expected because the production depth is inferred from $\Delta 47$ temperatures. Consequently, the production depth temperatures at adjacent core-top sites are close to $\Delta 47$ temperatures. Applying this $\Delta 47$ -UK'37 relationship to the North Atlantic Miocene UK'37 record will thus also yield temperatures that are similar to $\Delta 47$ temperatures. This key point needs to be addressed and clarified in the text.

The discussion assumes that UK'37 values reflect the temperatures during the months of coccolithophore production. However, this relationship is not straightforward in the field, due to substantial degradation during sinking and taphonomic processes, both of which can alter UK'37 values and distort alkenone production seasonality. Additionally, non-thermal

physiological factors, such as nutrient stress, can also bias UK'37 values. I encourage the authors to consider these points to strengthen their discussion of the potential causes of the warm bias in UK'37 temperatures before concluding that SST calibrations are the culprit. That said, I agree with the authors that the consistent offsets between $\Delta 47$ and UK'37 temperatures are likely due to systematic proxy issues, such as calibration.

I suggest that the section "Could alkenone records overestimate North Atlantic temperatures?" be restructured slightly by first discussing the culture calibrations before stating that the warm-biased UK'37 temperatures are due to the application of core-top SST calibrations. I wrote the line-by-line comments while going over the manuscript. As the authors will see below, I found myself thinking several times, "but the SST calibration agrees with the PrahI culture calibration, suggesting that these are production temperatures!"

Specific comments

Line 49: "however" does not sound right given the previous sentence. I suggest omitting it or rewording.

Line 80: remote sensed → remotely sensed

Line 106: "Negligible cold bias" In this section, the authors discuss whether the lower $\Delta 47$ temperatures are a result of proxy issues. As such, the authors may want to rephrase it to something along the lines of "Assessing the fidelity/potential caveats/sources of cold bias."

Line 116–120: The "cold" bias discussed here is relative to UK'37 temperatures. Laterally advected coccoliths from more subpolar areas would also affect UK'37, as alkenones are just as prone to advection, if not more so, than coccoliths. This possibility should be considered in the discussion.

Line 153–155: The same results would be obtained if they used the widely applied culture calibration from PrahI et al. (1988), which is identical to the calibration by Müller et al. (1998). This is often cited as a reason to support the assumption that sedimentary UK'37 reflects sea surface temperature (SST). However, the possibility that the agreement between the calibrations of PrahI et al. and Müller et al. is merely a coincidence (for UK'37 users, anyway) cannot be completely ruled out. Regardless, this raises a challenge to the main conclusion of the study, which posits that calibrating UK'37 to SST results in the overestimation of ocean temperature estimates. Given its relevance, including this consideration in the discussion would enhance the nuance and comprehensiveness of the arguments.

Line 157–162: Refer to my comments above regarding the similarity between the culture and core-top calibrations. Additionally, a more appropriate comparison with the $\Delta 47$ coccolith calcification temperature would, at least in principle, involve applying the UK'37 calibration based on production temperatures proposed by Conte et al. (2006) G-cubed. I encourage the authors to investigate whether this results in a better match with their $\Delta 47$ temperatures.

Line 160: surficial calibration → surface sediment/core-top calibration

Line 184–197: Please refer to my comments above on Lines 153–155 and 158–162. I suggest moving this paragraph to before Line 153, i.e., prior to concluding that the overestimation of UK'37 temperatures is due to the application of SST calibrations.

Differences in the culture studies may partly arise from the culturing methods (batch vs. continuous).

Line 198–208: Here, the authors argue that the better match with $\Delta 47$ temperatures when applying the calibration from Mejia et al. (2023) is an indication that the problem lies with the UK'37 SST calibration. I contend that this reasoning is flawed. In Mejia et al. (2023), the authors determine the alkenone production depths by matching the $\Delta 47$ temperatures of the samples with the T-depth profile from WOA. They then produce a calibration using the temperatures at these production depths. Although the regression is conducted on nearby core-top UK'37 values from Tierney and Tingley (2015), rather than the samples on which $\Delta 47$ measurements were conducted, the T-depth profiles at these sites—and thus the temperatures at production depths—are likely similar to those at the study sites of Mejia et al. (2023). Consequently, this approach effectively calibrates UK'37 values to $\Delta 47$ temperatures. Therefore, it is not surprising that applying this calibration to the Miocene record would yield similar temperature estimates to the $\Delta 47$ record.

Line 222–233: As it stands, the rather inconclusive discussion on TEX86 data in this paragraph does not add much to the debate on UK'37 vs. $\Delta 47$.

The authors could explore further. What are the GDGT 2/3 ratios from this site? Do they indicate subsurface or surface origin for sedimentary TEX86? The authors seem to interpret TEX86 as a shallow subsurface proxy, as they applied the subsurface TEX86 calibration from Tierney and Tingley (2015). The fact that the subsurface temperatures (0–200 m) derived from TEX86 better agree with UK'37 SSTs would, at face value, suggest that the UK'37 proxy is underestimating SSTs, which implies that $\Delta 47$ temperatures are substantially underestimated since the production depth of coccolithophores is within the same depth range as the TEX86 subT calibration (0–200 m). Have the authors tried other subsurface calibrations for TEX86, such as the ones from Kim et al. (2012) EPSL and Ho and Laepple (2016), to see how the resulting temperatures compare to UK'37 and coccolith $\Delta 47$?

Line 249: Typo “interprations”

Line 271: Typo “latiutude”

Line 275: Typo “probem”

Line 293: latitude → latitudinal

Line 299: One possible reason for the higher simulated EEP SSTs is that, generally speaking, models need improvements in simulating upwelling.

Line 314: Typo “attempts”

Line 323: “reproduced in other high latitudes” does not sound right. Please rephrase.

Line 325: Typo “anthropogeic”

Line 364–365: typo “at 2 at 10 μm ”

Line 395: Typo “PoraPakQ”

Line 402: “as considered affected” some words are missing.

Line 406 and 409: “culture coccolith $\Delta 47$ ” or “coccolith culture $\Delta 47$ ”

Line 421: Typo: “gran”

Figure 2: Perhaps mark the time intervals mentioned in the text, e.g., Mid-Miocene, Quaternary, etc., so that non-expert readers can better follow the result description and discussion.

Figure 5: Proxy data need error estimates.

Reviewer #2

(Remarks to the Author)

The issue of polar amplification is one of the great conundrums of the Miocene, and this manuscript highlights a problem with proxy data calibration that could address it, which would be very impactful. Overall, this is a nice manuscript with sound logic that tells a simple yet compelling story. Everything is well laid out and several times I found myself writing down a question only to have it answered in the next paragraph. Though I am not an expert in D47 or Uk37, the authors make a convincing case that D47 more accurately represents calcification temperature than Uk37, and that when season and calcification depth are accounted for, both D47- and Uk37-derived temperatures give more accurate representations of calcification temperature.

One area that is key to the major conclusions of the manuscript is the conversion of Miocene calcification temperatures to mean annual SST, which can then be compared to models to address meridional temperature gradients and polar amplification. It seems that the authors correct their D47 data for seasonal preference but not for the fact that coccoliths do not calcify directly at the surface. Adding this correction would shift estimated sea surface temperatures warmer, which would affect the major conclusion of the paper. Since the authors make such a convincing case for this effect being significant (e.g., lines 174-177) it seems like it should be done before temperatures can be compared to models. It also seems odd that applying a seasonal/depth calibration to the Uk37 data shifts the temperature estimate to ~9C colder (Fig. 4), whereas doing the reverse and estimating MAT from D47 shifts the temperature estimate only ~1C warmer (Fig. 5). Can the authors explain this? I would be interested to see a version of Fig. 5 where the temperature estimates for each proxy is broken out for each site/time into 1) actual calcification temperatures accounting for seasonal/depth preferences and 2) MAT equivalent.

There are also a couple of instances where there is an opportunity to compare to existing data. These are explained below, along with minor comments relating to specific lines of the manuscript.

Lines 174-177: How are the SSTs during season of production known for the mid-Holocene?

Lines 206-208: How about the Herbert et al. (2016) Uk37 data? Here and in Fig. 4.

Lines 265-266: Leutert et al. (2020) and Hou et al. (2023, *Climate of the Past*) have Miocene D47 records from the Southern Ocean, which I think should be included in the discussion of meridional temperature gradients and Figure 5. Though they are in the Pacific sector and not the Atlantic, the authors choose an EEP site for their equatorial temperatures so it seems like this discussion is global.

Lines 292-293: Can the authors explain why alkenones would represent well temperatures in the EEP? Though the seasonal temperature swings are less at low latitude, this area has a strong seasonal upwelling cycle with changes in

productivity, SST, and the depth of the mixed layer. The authors have made a good case for alkenones being affected by seasonality and depth, which I would think would be a problem in the EEP.

Line 365: I think you mean “at 2 to 10 μm ”?

Figure 1: Caption says red star but it is black. I suggest adding a mark indicating the 6.74C modern temperature corresponding to the Icelandic data.

Figure 2: The Uk37 all batch culture symbol in the legend is invisible for me.

Figure 5: When I think of meridional temperature gradients, I think of Figure 12 from Burls et al. (2021). Since the authors are comparing directly to that model, it would be very compelling to show the alignment of temperatures calculated herein in the context of that type of figure. This is just a suggestion, but I think it would be helpful for the readers and make a more compelling case to show the Burls curves somewhere.

Supplementary Methods oceanographic setting: Since so much of this work relates to seasonality and the authors make a conversion from seasonal to annual temperature, it would be helpful to see some numbers on the seasonal swings in SST and even euphotic zone temperature, if available.

Version 1:

Reviewer comments:

Reviewer #1

(Remarks to the Author)

I would like to thank the authors for their detailed responses to my comments on the previous version. I find the manuscript improved following the revisions, and I am pleased to recommend publication pending the following minor editorial revisions.

L110: Suggested revision of subheading “Assessing the fidelity of coccolith $\Delta 47$ temperatures: negligible evidence for a cold bias”

L163–165: Consider citing a more comprehensive review on UK’37 calibration (including cultures) by Herbert, 2001 G-cubed (<https://doi.org/10.1029/2000GC000055>)

L350: Bayspline  BAYSPLINE

L384: “...publicly includes...” reads awkward. Please rephrase for clarity.

L386: Suggested revision “...differ by 0.9 $^{\circ}\text{C}$ from the mean annual temperature of the upper 0–70 m water column...”

L482: PoraPak  Porapak (this is a registered trademark)

Finally, I would like to congratulate the authors on this excellent piece of work, which provides valuable insights and challenges our current understanding of the widely applied UK’37 proxy and the latitudinal temperature gradient during the warm Miocene.

Reviewer #2

(Remarks to the Author)

I am satisfied with the authors’ responses. Since, as the authors point out, this manuscript is more focused on the potential of the clumped isotope proxy, and that is not my area of expertise, I leave that to Reviewer 1. Below are a few minor things I noticed from the parts of the manuscript related to my review.

Figure 4: The “a” label is a bit confusing, perhaps move it either up to the top of the box or down to the top of the data.

Line 303: I’m not sure what you mean by magnitude here, a few more words are needed.

Line 325: “could be invalid”

Line 759: “from which it differs by”

Line 760: “between 4 and 4.5” otherwise use a dash after 2.8

Supplement line 327: “The only”

We thank the reviewer for their constructive comments, which allowed us to improve our manuscript. Please find below our response to specific comments in blue, and in *blue italics*, the extracts of the manuscript text that relate to those comments. Lines refer to the finalized text files (no track changes).

REVIEWER COMMENTS

Reviewer #1 (Remarks to the Author):

1. In this study, the authors generate a North Atlantic temperature record spanning the past 16 million years using coccolith $\Delta 47$. This approach is relatively novel, and $\Delta 47$ is theoretically a proxy that is not affected by issues plaguing other proxies, making it a superior choice. The coccolith $\Delta 47$ temperature record is approximately 9°C lower than that of alkenone-based UK'37. Since both coccoliths and alkenones are produced by coccolithophores, these proxies should yield similar temperature estimates. The authors suggest that the discrepancy between the proxies is due to issues with UK'37 calibrations. Additionally, the coccolith $\Delta 47$ proxy outperforms commonly applied UK'37 and TEX86 in terms of agreement with climate models. $\Delta 47$ temperatures support a modest decrease in the latitudinal temperature gradient during the Miocene, in contrast to the extreme gradient flattening suggested by UK'37 and TEX86.

Despite numerous typos, the manuscript is well-written and enjoyable to read. This is one of the first reconstruction studies to use coccolith $\Delta 47$, scoring high on the novelty scale. Compared to other commonly used temperature proxies, $\Delta 47$ is not yet known to be affected by non-thermal issues that undermine the reliability of other proxies. As is common in proxy development, more issues may be uncovered later in its development stage. However, at this moment, coccolith $\Delta 47$ is a superior proxy, lending credibility to the resulting reconstructions. If the conclusion that UK'37 temperatures are severely overestimated proves to be true, this would have broad implications for paleoceanography across different timescales, given the widespread use of UK'37 in the community.

While I find the study timely and the results important, I am not yet fully convinced by some of the arguments presented. Some reasoning appears incomplete and requires further evidence and clarification. I hope the line-by-line comments will be helpful in strengthening these arguments when the authors revise the manuscript. The methods section is clear and sufficiently detailed for the work to be reproduced, but proxy errors should be considered in the model-data comparison.

The main finding of the study is that $\Delta 47$ temperatures are substantially lower than UK'37 temperatures. The authors did a great job at carefully considering and then ruling out potential cold bias in coccolith $\Delta 47$ data. By contrast, the discussion of UK'37 data is less in depth, drawing heavily from Mejia et al. (2023), which proposes that the application of UK'37 SST calibrations leads to temperature overestimation, as the proxy signal originates from subsurface waters in the tropics. Mejia et al. (2023) then generated a UK'37 production temperature calibration using subsurface temperatures (150m) in the tropics and SST in high latitudes, which naturally differs in sensitivity compared to those widely applied SST calibrations (i.e., SST latitudinal gradient > gradient between low latitude subsurface and high latitude SST). The authors then applied Mejia et al. (2023) UK'37 production temperature calibration to their Miocene record, which improves the match with the $\Delta 47$ record. However, this is to be expected because the production depth is inferred from $\Delta 47$ temperatures. Consequently, the production depth temperatures at adjacent core-top sites are close to $\Delta 47$ temperatures. Applying this $\Delta 47$ -UK'37 relationship to the North Atlantic Miocene UK'37 record will thus also yield temperatures that are similar to $\Delta 47$ temperatures. This key point needs to be addressed and clarified in the text.

The discussion assumes that UK'37 values reflect the temperatures during the months of coccolithophore production. However, this relationship is not straightforward in the field, due to substantial degradation during sinking and taphonomic processes, both of which can alter UK'37 values and distort alkenone production seasonality. Additionally, non-thermal physiological factors, such as nutrient stress, can also bias UK'37 values. I encourage the authors to consider these points to strengthen their discussion of the potential causes of the warm bias in UK'37 temperatures before concluding that SST calibrations are the culprit. That said, I agree with the authors that the consistent offsets between $\Delta 47$ and UK'37 temperatures are likely due to systematic proxy issues, such as calibration.

I suggest that the section "Could alkenone records overestimate North Atlantic temperatures?" be restructured slightly by first discussing the culture calibrations before stating that the warm-biased

UK'37 temperatures are due to the application of core-top SST calibrations. I wrote the line-by-line comments while going over the manuscript. As the authors will see below, I found myself thinking several times, "but the SST calibration agrees with the Prahl culture calibration, suggesting that these are production temperatures!"

As suggested by the reviewer, we have modified the section: "Could alkenone records overestimate North Atlantic temperatures?" (L149). We have now organized this section with the culture calibration part in the beginning. In this part, we have now directly stated that the only culture calibration that coincides with core top and Bayspline calibrations is the one of Prahl, which is specific for a batch culture of a certain strain of *E. huxleyi*. We have shown that other culture calibrations (compiled by d'Andrea et al., 2016) yield temperatures that are significantly different from each other and stated that non-thermal effects might contribute to explain these, as proposed by D'Andrea (2016). We have also mentioned some of the non-thermal effects suggested by the reviewer, in addition to growth phase, light availability or unique physiology of species/strains. We have also included other potential alkenone biasing effects like preferential degradation and lateral transport, as suggested by the reviewer (L165-174). The first part of this section now reads:

*"If alkenones were produced under the same conditions (i.e. season, depth, light, nutrients, growth phase) during which coccolithophores calcify, we would expect similar absolute temperature estimates from both proxies. However, despite sharing similar trends and being correlated (Fig. 2, Supplementary Fig. 6), coccolith Δ_{47} calcification temperatures are significantly colder than alkenone temperatures estimated using calibrations based on regressions of $U_{37}^{k'}$ against SSTs^{20,21}. From existing alkenone culture calibrations (analog to Δ_{47} calibrations), only the batch *Emiliana huxleyi* (strain 55a) calibration²² agrees with calibrations based on SSTs^{20,21}. Several other culture studies on different strains of *E. huxleyi* and *Gephyrocapsa oceanica* show different alkenone unsaturation calibrations to growth temperatures³¹. When applied to our $U_{37}^{k'}$ dataset, there are up to 8.0 °C differences among these culture experiments and surprisingly, all yield even warmer temperatures than the modern SST and the published coccolith Δ_{47} core top temperature¹⁷ (Fig. 2a, Supplementary Table 5). Similarly, the application of the alkenone calibration based on field-measured temperatures³² to $U_{37}^{k'}$ of the Holocene sample¹⁷ result in even larger (>7 °C) overestimates of modern SSTs (Fig. 2a). These large differences in the sensitivity of $U_{37}^{k'}$ to cultured temperature highlight the importance of understanding potential non-thermal effects in alkenones, such as nutrient stress and the related cellular growth phase (exponential, stationary), light availability, type of culture method (batch or continuous), or physiological aspects unique to a species or strain^{33,34}. In addition to non-thermal effects, $U_{37}^{k'}$ measured on fossil samples can be affected by preferential degradation of $C_{37:3}$ in highly oxygenated waters³⁵⁻³⁷, which could be particularly important for the North Atlantic. Moreover, temperature biases could also arise from laterally-transported alkenones attached to coarse easily-transported sediment fractions³⁸ potentially produced in warmer areas. Both mechanisms could introduce a warm bias in our alkenone-derived temperatures. Improving our knowledge on aspects like the utility of synthesizing alkenones, cellular production pathways, and all possible non-thermal mechanisms would help clarifying absolute temperatures (Fig. 2a, b) and which calibrations are most appropriate for a given oceanographic setting."*

Despite all these potential effects, as the reviewer points out, the differences between both proxies are in general consistent (trends are very similar), which points to a rather more "consistent" source of differences, like calibrations, to explain the offset in absolute temperatures. We completely agree with the reviewer on the importance of mentioning non-thermal effects, alkenone degradation and lateral transport as potential biasing factors, so we have now included them in the discussion. Some of these factors, however, are likely to have varied temporally. For instance, whether alkenones are being produced by one species or strain, nutrient depleted or not, with light limitation or not, at a specific growth rate or growth phase, etc. These are all non-thermal factors that if found to be important, we would expect less "consistent" differences between clumped isotopes and alkenones. Therefore, we retain the hypothesis that at least for this Site and time interval, calibration approaches may explain a relatively large part of relatively consistent offset in absolute temperatures (~9 °C).

The recalibration of alkenones based on depth and season of production suggested by coccolith core top clumped isotopes (Mejia et al., 2023) is just a "rough" approach not intended to replace a statistically-well constrained alkenone calibration. To be applied with confidence, the dataset should be extensively amplified in terms of data and geographic locations. The idea behind it is that since we are confident that clumped isotopes lead to coccolithophore's calcification temperatures, these should be an indicator of their habitat depth and therefore, better explain their ecology compared to the

alkenone interpretation of an unvarying surficial habitat. We do not agree with the reviewer that applying this calibration leads to the flawed argument that alkenone vs. clumped calibrations could explain a large part of magnitude differences between our proxies in the North Atlantic. The Mejia et al., 2023 alkenone calibration was conducted using alkenone data from a completely independent data set (core top from Tierney and Tingley 2018, Bayspline study) and from the season and depth of production temperatures suggested by clumped isotopes from another independent dataset (core top from Mejia et al., 2023). The Mejia et al., 2023 calibration assumes that alkenones are being produced during the main season of coccolithophore production and at the habitat depth suggested by core top clumped isotope temperatures, instead of during August-October and at 0 m for the North Atlantic and mean annual at 0 m for other parts of the world. Since these datasets are independent from our new Miocene record, it should be possible to apply the alternative alkenone calibration to our independent North Atlantic alkenone data, without necessarily expecting temperatures to agree with our clumped temperatures over 16 My. Moreover, core top calibrations are meant to represent as many oceanographic areas and settings as possible, so that when applied to an independent dataset, variability linked to geographical settings are accounted for. Therefore, instead of seeing the geographical “match” between our North Atlantic Site and the coretop independent dataset as a con, we think it is a pro, as it includes in the dataset that best matches the ocean characteristics of the North Atlantic, despite of the calibration still being “rough”. This discussion and our results here presented are intended to represent a starting point and hopefully will stimulate greater work in the community to integrate these discrepancies further, produce an improved alkenone calibration, leading to more precise temperature reconstructions.

Specific comments

2. Line 49: “however” does not sound right given the previous sentence. I suggest omitting it or rewording.

Eliminated, as suggested by reviewer.

3. Line 80: remote sensed → remotely sensed

Modified, as suggested by reviewer.

4. Line 106: “Negligible cold bias” In this section, the authors discuss whether the lower Δ_{47} temperatures are a result of proxy issues. As such, the authors may want to rephrase it to something along the lines of “Assessing the fidelity/potential caveats/sources of cold bias.”

We have decided to include the reviewer’s suggestion while keeping the end message of the original section title. In general, we like to start sections with a summary and assertive sentence of what the paragraph is going to conclude. Therefore, we would like to keep the message that cold biases are negligible. As suggested by the reviewer, we now include in the section title: “*Assessing fidelity of the coccolith Δ_{47} record: a cold bias is negligible*”

5. Line 116–120: The “cold” bias discussed here is relative to UK’37 temperatures. Laterally advected coccoliths from more subpolar areas would also affect UK’37, as alkenones are just as prone to advection, if not more so, than coccoliths. This possibility should be considered in the discussion.

The lateral advection argument has been now added to section: “Could alkenone records overestimate North Atlantic temperatures?”, as suggested by the reviewer (L168-174). The sentence now reads:

“In addition to non-thermal effects, U_{37}^k measured on fossil samples can be affected by preferential degradation of $C_{37:3}$ in highly oxygenated waters^{35–37}, which could be particularly important for the North Atlantic. Moreover, temperature biases could also arise from laterally-transported alkenones attached to coarse easily-transported sediment fractions³⁸ potentially produced in warmer areas. Both mechanisms could introduce a warm bias in our alkenone-derived temperatures.”

6. Line 153–155: The same results would be obtained if they used the widely applied culture calibration from Prah et al. (1988), which is identical to the calibration by Müller et al. (1998). This is often cited as a reason to support the assumption that sedimentary UK’37 reflects sea surface temperature (SST). However, the possibility that the agreement between the calibrations of Prah et al. and Müller et al. is merely a coincidence (for UK’37 users, anyway) cannot be completely ruled out.

Regardless, this raises a challenge to the main conclusion of the study, which posits that calibrating UK'37 to SST results in the overestimation of ocean temperature estimates. Given its relevance, including this consideration in the discussion would enhance the nuance and comprehensiveness of the arguments.

The discussion of the culture calibrations has been now moved to the beginning of the section, as suggested by the reviewer. Here, we point out directly that only the culture calibration of Prahl coincide with the widely used coretop calibrations, while other culture calibrations show significant differences (see d'Andrea et al., 2016). We also add a discussion of different non-thermal effects that could explain such differences between culture calibrations (please also refer to response to reviewer point 1). This part of the section now reads:

*“From existing alkenone culture calibrations (analog to Δ_{47} calibrations), only the batch *Emiliana huxleyi* (strain 55a) calibration²² agrees with calibrations based on SSTs^{20,21}. Several other culture studies on different strains of *E. huxleyi* and *Gephyrocapsa oceanica* show different alkenone unsaturation calibrations to growth temperatures³¹. When applied to our $U_{37}^{k'}$ dataset, there are up to 8.0 °C differences among these culture experiments and surprisingly, all yield even warmer temperatures than the modern SST and the published coccolith Δ_{47} core top temperature¹⁷ (Fig. 2a, Supplementary Table 5). Similarly, the application of the alkenone calibration based on field-measured temperatures³² to $U_{37}^{k'}$ of the Holocene sample¹⁷ result in even larger (>7 °C) overestimates of modern SSTs (Fig. 2a). These large differences in the sensitivity of $U_{37}^{k'}$ to cultured temperature highlight the importance of understanding potential non-thermal effects in alkenones, such as nutrient stress and the related cellular growth phase (exponential, stationary), light availability, type of culture method (batch or continuous), or physiological aspects unique to a species or strain^{33,34}.”*

7. Line 157–162: Refer to my comments above regarding the similarity between the culture and core-top calibrations. Additionally, a more appropriate comparison with the Δ_{47} coccolith calcification temperature would, at least in principle, involve applying the UK'37 calibration based on production temperatures proposed by Conte et al. (2006) G-cubed. I encourage the authors to investigate whether this results in a better match with their Δ_{47} temperatures.

Please refer to response to reviewer point 1 regarding the culture calibrations. In this version of the manuscript we have included alkenone temperature calculations applying the field surface water calibration from Conte et al., (2006), as suggested by the reviewer. The results are now shown in Fig 2 of the main text. We show only the global equation, as the one specific for the Atlantic yields to similar results and we do not want to overcrowd the figure. Temperatures using this calibration are generally similar to calibrations based on SSTs (coretops), except for the lowest temperatures (youngest samples) for which temperatures are even warmer (as explained in Conte's et al paper). This calibration results in alkenone temperatures for the Holocene sample that are much higher than modern mean annual SSTs (>7 °C) or season of production SSTs (>8 °C). In the manuscript we connect the results using this calibration and those of cultures with potential non-thermal effects, alkenone degradation and lateral transport, as suggested by the reviewer (L 158-174).

The new sentences of the main text now read: *“...Several other culture studies on different strains of *E. huxleyi* and *Gephyrocapsa oceanica* show different alkenone unsaturation calibrations to growth temperatures³¹. When applied to our $U_{37}^{k'}$ dataset, there are up to 8.0 °C differences among these culture experiments and surprisingly, all yield even warmer temperatures than the modern SST and the published coccolith Δ_{47} core top temperature¹⁷ (Fig. 2a, Supplementary Table 5). Similarly, the application of the alkenone calibration based on field-measured temperatures³² to $U_{37}^{k'}$ of the Holocene sample¹⁷ result in even larger (>7 °C) overestimates of modern SSTs (Fig. 2a). These large differences in the sensitivity of $U_{37}^{k'}$ to cultured temperature highlight the importance of understanding potential non-thermal effects in alkenones, such as nutrient stress and the related cellular growth phase (exponential, stationary), light availability, type of culture method (batch or continuous), or physiological aspects unique to a species or strain^{33,34}. In addition to non-thermal effects, $U_{37}^{k'}$ measured on fossil samples can be affected by preferential degradation of $C_{37:3}$ in highly oxygenated waters^{35–37}, which could be particularly important for the North Atlantic. Moreover, temperature biases could also arise from laterally-transported alkenones attached to coarse easily-transported sediment fractions³⁸ potentially produced in warmer areas. Both mechanisms could introduce a warm bias in our alkenone-derived temperatures.”*

8. Line 160: surficial calibration → surface sediment/core-top calibration

In previous round of reviews we have introduced this word “surficial” calibration to refer to calibrations using sea surface temperatures. Now here we have changed this word, since it seems to be confusing, to state directly the coretop and Bayspline calibrations. The sentence now reads:

“Consequently, in places or time intervals in which coccolith biomineralization (and alkenone formation) occurs at depth and/or during cooler seasons, the application of coretop²⁰ and Bayspline²¹ calibrations to $U_{37}^{k'}$ records is expected to overestimate mean annual SSTs and produce warmer estimates than actual calcification temperatures derived from Δ_{47} .”

9. Line 184–197: Please refer to my comments above on Lines 153–155 and 158–162. I suggest moving this paragraph to before Line 153, i.e., prior to concluding that the overestimation of UK'37 temperatures is due to the application of SST calibrations. Differences in the culture studies may partly arise from the culturing methods (batch vs. continuous).

As suggested by the reviewer, we have moved this paragraph to the beginning of the section and deepened the non-thermal effect arguments and other effects like degradation and lateral transport (and batch vs continuous cultures as well).

10. Line 198–208: Here, the authors argue that the better match with Δ_{47} temperatures when applying the calibration from Mejia et al. (2023) is an indication that the problem lies with the UK'37 SST calibration. I contend that this reasoning is flawed. In Mejia et al. (2023), the authors determine the alkenone production depths by matching the Δ_{47} temperatures of the samples with the T-depth profile from WOA. They then produce a calibration using the temperatures at these production depths. Although the regression is conducted on nearby core-top UK'37 values from Tierney and Tingley (2015), rather than the samples on which Δ_{47} measurements were conducted, the T-depth profiles at these sites—and thus the temperatures at production depths—are likely similar to those at the study sites of Mejia et al. (2023). Consequently, this approach effectively calibrates UK'37 values to Δ_{47} temperatures. Therefore, it is not surprising that applying this calibration to the Miocene record would yield similar temperature estimates to the Δ_{47} record.

Please refer to response to response to point 1 of reviewer for a detailed answer. In any case, we also want to point out that because it has been shown that coccolith Δ_{47} is thermodynamically controlled with laboratory cultures, sediment traps and coretops (Clark et al. 2024, Clark et al. 2025, Mejia et al. 2023), the fact that the UK 37 temperatures calculated with this calibration are compatible with the Δ_{47} record is not merely an artifact of the calibration of Uk37 to Δ_{47} in Mejia et al (2023), but it's a realistic representation of the temperatures at which the alkenones have been synthesized.

11. Line 222–233: As it stands, the rather inconclusive discussion on TEX86 data in this paragraph does not add much to the debate on UK'37 vs. Δ_{47} .

The authors could explore further. What are the GDGT 2/3 ratios from this site? Do they indicate subsurface or surface origin for sedimentary TEX86? The authors seem to interpret TEX86 as a shallow subsurface proxy, as they applied the subsurface TEX86 calibration from Tierney and Tingley (2015). The fact that the subsurface temperatures (0-200 m) derived from TEX86 better agree with UK'37 SSTs would, at face value, suggest that the UK'37 proxy is underestimating SSTs, which implies that Δ_{47} temperatures are substantially underestimated since the production depth of coccolithophores is within the same depth range as the TEX86 subT calibration (0-200 m). Have the authors tried other subsurface calibrations for TEX86, such as the ones from Kim et al. (2012) EPSL and Ho and Laepple (2016), to see how the resulting temperatures compare to UK'37 and coccolith Δ_{47} ?

We thank the reviewer for getting our attention back to TEX₈₆. Although we are convinced that to conduct appropriate comparisons to TEX₈₆ we would need a deep analysis of calibrations, ecology, seasonality and depth of production of archaea in our site, as we did for coccolithophores, which is beyond the scope of this paper, in this version of the manuscript we applied the Ho and Laepple subsurface calibration and added the results to the supplementary figure, and now discussed them in the main text. Here we clarify that in the previous version of this manuscript we did not attempt to interpret TEX₈₆ as coming from any specific depth (or calibration), as we simply reported the data from the original paper of Super et al., 2020, who calculated temperatures both using the Bayspar and the TEX₈₆H calibrations (but show the Bayspar calibration results). While doing these calculations and

revising the figure, we realized that in the previous version of supplementary figure 7, we have plotted Bayspar temperatures from Site 608 but TEX₈₆h temperatures from 982 (Kim et al calibration). In this version, we have now plotted both Bayspar calibrations for both sites from Super et al., 2020, and added the additional results when we apply the subsurface calibration of Ho and Laepple. TEX₈₆ temperatures using this calibration are generally colder than coccolith clumped isotope temperatures, while not being extremely colder (Supplementary Fig. 7). For a Site with a very small thermocline (during production season) like ODP 982 and only 0.36 C difference between winter-spring SST and winter-spring average 0-550 m temperatures, this result is not surprising and fits with our coccolith clumped interpretations. The new paragraph reads:

“Miocene TEX₈₆ absolute temperatures for ODP Site 982 calculated using a shallow subsurface calibration⁴⁵ show generally similar (or slightly colder) values compared to a location 14.7° further south in the subtropical gyre⁸, and generally fall between our alkenone and coccolith Δ_{47} -derived temperature estimates (Supplementary Fig. 7). Similar temperatures at ODP Site 982 and at the subtropical gyre would be possible under extreme high latitude amplification. Alternatively, it is also possible that there was still a latitudinal thermal gradient between these sites that cannot be discerned by TEX₈₆ at these locations or time intervals, potentially due to similar challenges in the attribution of the production depth and season. If accurate, these TEX₈₆ “warm subsurface” values would indicate that both alkenone and coccolith Δ_{47} temperatures underestimate euphotic zone temperatures. However, when we apply the subsurface TEX₈₆ calibration⁴⁶ to this dataset⁸, absolute values mostly fall under those suggested by coccolith Δ_{47} temperatures (Supplementary Fig. 7). Considering that the subsurface calibration⁴⁶ has its highest occurrence at integrated 0-550 m water depths, and that the difference between SSTs and mean 0-550 m temperatures in the modern ocean during the winter-spring season is very small (~0.36° C) due to the small thermocline, slightly lower but similar TEX₈₆ estimates compared to the euphotic zone (~71 m) coccolith Δ_{47} temperatures are expected (Supplementary Fig. 7). While this discussion provides some context for TEX₈₆ temperature comparisons to our proxies, we highlight that alkenone and coccolith Δ_{47} temperatures can be readily compared because they derive from the same organism. Detailed comparison of absolute coccolith Δ_{47} temperatures with other records from proxies based on other organisms, like TEX₈₆, would require a thorough analysis not only of calibrations but also poorly constrained differences in the ecology of the biomarker producing archaea which is beyond the scope of this paper.”

12. Line 249: Typo “interprations”
Corrected.

13. Line 271: Typo “latiutude”
Corrected.

14. Line 275: Typo “probem”
Corrected

15. Line 293: latitude → latitudinal
Corrected

16. Line 299: One possible reason for the higher simulated EEP SSTs is that, generally speaking, models need improvements in simulating upwelling.

This has been now added to the sentence as suggested by reviewer (L 340).

17. Line 314: Typo “attempts”
Corrected.

18. Line 323: “reproduced in other high latitudes” does not sound right. Please rephrase.

We have now modified the sentence to: “If the more modest Mid and Late Miocene high latitude warmth shown by our coccolith Δ_{47} is observed in other high latitude sites,…” (L 369)

19. Line 325: Typo “anthropogeic”
Corrected.

20. Line 364–365: typo “at 2 at 10 μ m”

Corrected.

21. Line 395: Typo "PoraPakQ"

We now write PoraPak Q. This is not a typo.

22. Line 402: "as considered affected" some words are missing.

Now modified to: *"Measurements with Δ_{48} offset > 2 and 49 parameter values > 2 ‰ were eliminated because it was considered that they were affected by contamination⁶⁶" (L 448)*

23. Line 406 and 409: "culture coccolith Δ_{47} " or "coccolith culture Δ_{47} "

Both now as coccolith culture

24. Line 421: Typo: "gran"

Modified

25. Figure 2: Perhaps mark the time intervals mentioned in the text, e.g., Mid-Miocene, Quaternary, etc., so that non-expert readers can better follow the result description and discussion.

Pleistocene, Pliocene, Late and Mid Miocene now highlighted in this and all other figures as suggested by reviewer

26. Figure 5: Proxy data need error estimates.

We have now redone Fig 5 to include not only modelled mean annual temperatures from Site 982 and the EEP, but all latitudes between 0-80 N, which are relevant in our discussion. We have done this because reviewer 2 suggested to present model vs. proxy data in this format, which is the standard way of model vs. proxy comparisons.

In previous revisions of this manuscript, other reviewers have asked for simplification of this figure, therefore, we have decided to not include errors to improve readability of this figure in the main text. However, we acknowledge it is important to show errors as well, so we have now included an additional supplementary figure (Suppl. Fig 9) that includes the errors of both our alkenone (Bayspline calibration) and our clumped isotope temperatures. In addition, we have now stated the broad error estimates in the figure caption of the main text. We are open to switch this supplementary figure to the main text if after looking at both figures, the reviewers and editor think including these errors do not compromise readability in a later revision of the manuscript.

Reviewer #2 (Remarks to the Author):

The issue of polar amplification is one of the great conundrums of the Miocene, and this manuscript highlights a problem with proxy data calibration that could address it, which would be very impactful. Overall, this is a nice manuscript with sound logic that tells a simple yet compelling story. Everything is well laid out and several times I found myself writing down a question only to have it answered in the next paragraph. Though I am not an expert in D47 or Uk37, the authors make a convincing case that D47 more accurately represents calcification temperature than Uk37, and that when season and calcification depth are accounted for, both D47- and Uk37-derived temperatures give more accurate representations of calcification temperature.

1. One area that is key to the major conclusions of the manuscript is the conversion of Miocene calcification temperatures to mean annual SST, which can then be compared to models to address meridional temperature gradients and polar amplification. It seems that the authors correct their D47 data for seasonal preference but not for the fact that coccoliths do not calcify directly at the surface. Adding this correction would shift estimated sea surface temperatures warmer, which would affect the major conclusion of the paper. Since the authors make such a convincing case for this effect being significant (e.g., lines 174-177) it seems like it should be done before temperatures can be compared to models. It also seems odd that applying a seasonal/depth calibration to the Uk37 data shifts the temperature estimate to ~9C colder (Fig. 4), whereas doing the reverse and estimating MAT from D47 shifts the temperature estimate only ~1C warmer (Fig. 5). Can the authors explain this? I would be interested to see a version of Fig. 5 where the temperature estimates for each proxy is broken out for

each site/time into 1) actual calcification temperatures accounting for seasonal/depth preferences and 2) MAT equivalent.

This paper is not envisioned to be only focused on model-data comparisons but rather show the great potential that clumped isotopes in coccoliths have to produce independent temperature records that accurately reconstruct both trends and absolute values. Therefore, we first attempted to compare mean annual temperatures from models with our coccolith clumped temperatures. But it is true for this Site, we hypothesize coccolith clumped temperatures are indicating winter-spring temperatures at production depth.

We agree with the reviewer that ideally model-data comparisons should be done considering any seasonal bias, and also depth bias, if any (or any other suspected bias). Unfortunately, studies focusing on model-data comparisons, which have discussed the conundrum of the Miocene and other high latitude warmth periods for many years, typically take temperatures directly from proxy publications and compare them to mean annual temperatures from models. In the best-case scenario, they apply the same calibration to all published proxy data and compare to modelled mean annual temperatures. This means, that any bias contained in a specific proxy or calibration (e.g. seasonal, or depth), is usually directly compared to modelled mean annual temperatures, making these comparisons not ideal. For instance, in the case of alkenone data derived from applying the widely-used Bayspline calibration, SSTs obtained will be indicators of August-October temperatures (at our site), which the calibration uses as its base. Therefore, model-data comparisons in which data are derived from alkenones using the Bayspline calibration would be comparing mean annual temperatures from models and August-October temperatures from proxies. The majority of modelled seasonal and water depth temperatures for our study interval are not publicly available. We now state this in the main text (L 314). Therefore, we have to use the traditional approach to compare modelled mean annual temperatures directly with proxy data that includes some season/depth bias. One exception is the model of Lee et al., recently published this year which includes mean annual temperatures of the water column. We have now included in the discussion, the supplementary information and the new Fig 5 both SSTs from this model and mean annual temperatures between 0-70 m (which is the limit of the euphotic zone for ODP 982 during the season of production). Modelled water column temperatures from this model do not, however, isolate a potential season of production effect.

An alternative approach of comparisons to models that we have already added to the previous version of the manuscript, was conducting a back calculation from coccolith clumped temperatures (winter-spring) to mean annual temperatures assuming the season of production remained the same since the Miocene to the modern (0.97 °C). The reviewer now asks us to include in this back calculation the effect of depth as well. In this version of the manuscript we have done this, again, assuming the depth effect remained constant through time as it is in the modern ocean (0.34 °C). These calculations are now explained in detail in the new Supplementary Note 5 (see below). We have also modified the main text accordingly (L 320-330).

Here we want to note, however, that assuming that the modern ocean conditions were valid for the North Atlantic since the Miocene is an approach to circumvent the lack of seasonal and depth modelled temperatures for comparison to our clumped temperatures, and such an assumption could be not valid for a climate state and continental configuration different to that of today. We are convinced the solution of back-calculating clumped isotope temperatures (or any other proxy with a bias dependent on ecological factors) to surficial mean annual temperatures is far from ideal. We cannot be certain of several controlling factors that likely varied in time like temperature contrast between seasons, changes in coccolithophore season and depth of production, mixing, nutrient availability and very importantly, changes in the relationship between SSTs and temperatures at depth at our site.

As the reviewer points out, applying the Mejia et al., (2023) calibration for alkenones which considers season and depth of production to our North Atlantic data, lowers down alkenone temperatures to magnitudes similar to those of clumped isotopes of coccoliths (so around 9 °C). But doing the back-calculation from coccolith clumped isotopes to surficial mean annual temperatures considering modern North Atlantic Ocean conditions only adds 1.31 °C to our coccolith clumped temperatures. This is because in the modern ocean, alkenone values are low (UK37=0.56 for the core top Holocene sample), and at this value the difference between traditional calibrations (e.g. see in green in the following figure; Muller core top calibration) and the Mejia et al., 2023 calibration (blue in figure) is not that large (15.5 °C for core top calibration; 10.9 °C for Mejia et al., 2023; modern mean annual SST

10.57 °C; modern season of production SST 9.6 °C). Note that alkenone temperatures of the most recent Holocene sample calculated with traditional calibrations already overestimate mean annual SSTs by almost 5 °C compared to modern ocean SSTs (see for instance Fig 2 and 3 of the main text). On the other hand, when alkenone values are higher (see again graph below), as it is the case for the Miocene samples (e.g. UK37=0.97 oldest sample), the temperature estimates using each calibration deviates much more (core top: 28 °C; Mejia et al., 2023: 19.2 °C). It is therefore expected that the application of the Mejia et al., 2023 calibration to older samples with higher values of UK37 would have a larger impact in decreasing alkenone temperatures compared to as if we consider potential season and depth biases of the modern ocean.

We do not know if the relationship between SST and temperatures at depth remained the same through time. We do not even know with certainty if oceans would have been more stratified in the Miocene (and will be in the future, as the IPCC projects) and perhaps high latitudes will be more similar to modern tropical oceans, or if there will be much more mixing due to increased energy producing storms or other mixing mechanisms (as the Miocene modelling paper of Lohman et al., 2022 suggests). Therefore, although we have made an attempt to back calculate coccolith clumped isotopes to surficial mean annual temperatures to compare to model outputs, as the reviewer suggested, which is already a step not conducted amongst most proxy-data comparisons, we are convinced the ideal way to do such comparisons would be comparing the modelled seasonal and depth temperatures to proxy data directly, as doing back calculations entails unquantifiable biases in surficial mean annual temperature estimates.

In blue: Mejia et al., 2023 calibration considering season and depth of production
 In green: Coretop calibration from Muller (based in mean annual SSTs).

The related paragraphs in the main text now read:

“The best fit with current proxy data for the Late Miocene was achieved with the NorESM-L model set at 560 ppm of CO₂ (higher than proxy estimates), which includes an improved representation of cloud microphysics and led to the best capture of polar amplification during the Eocene⁵⁵. Mean annual temperature Late Miocene simulations from this model show latitudinal temperature gradients between ODP Site 982 and the Eastern Equatorial Pacific (EEP) of ~14.7 °C, slightly smaller than those of the modern ocean (~15.4 °C). Ideally model-data comparisons should be based on the same season, and consider the depth from which the proxy temperature signal originates. However, the majority of modelled seasonal and water depth temperatures for the Miocene are not publicly available and comparisons have been traditionally made with proxy data without considering potential proxy seasonality or depth biases (e.g. North Atlantic alkenone-derived SST using the Bayspline calibration indicate August-October SSTs). Unfortunately, neither the winter-spring or the water column temperature structure from this model are publicly available. Therefore we take the traditional approach and directly compare model mean annual temperatures with coccolith Δ_{47} -derived winter-spring temperatures (productive season). In addition, we have attempted to back-calculate surface mean annual temperatures from our coccolith Δ_{47} to compare to modelled mean annual temperatures, assuming the difference between surface mean annual temperatures and winter-spring temperatures at production depth since the Miocene was similar to that of the modern ocean (seasonal effect = ~0.97 °C; depth effect = ~0.34 °C; total = 1.31 °C; Supplementary Note 5). While this assumption could be not valid in a changing ocean since the Miocene, given the different climate state and continental distribution, leading to likely varying temperature contrast between seasons, changes in coccolithophore season and depth of production, and changes in the relationship between SSTs and temperatures at depth at our site, it is the only way we can currently directly compare measured

coccolith Δ_{47} temperatures and modelled mean annual temperatures. Assuming that alkenones represent well temperatures of the EEP⁵⁶, latitudinal temperature gradients for the Late Miocene calculated using our North Atlantic coccolith Δ_{47} (~13.5-12.1 °C) are much more consistent with climate models than those derived from the average of all other proxies available for our location (~4.7 °C including our alkenone record) (Fig. 5, Supplementary Fig. 9). Although the traditional interpretation of alkenone calibrations to SSTs and non-thermal processes could lead to overestimates of EEP temperatures as well¹⁷, comparisons to high latitude temperatures does not change our conclusions of a better match of coccolith Δ_{47} calcification temperatures with models in the North Atlantic high latitudes.

For the Mid-Miocene, even CO₂ concentrations of 850 ppm were not able to reproduce the even flatter temperature gradient shown by proxy data (best fitting models: HadCM3L⁴), and EEP temperatures were overestimated by > 3 °C, which could be related to model limitations in simulating upwelling regions. Lower CO₂ concentrations improve tropical temperature simulations, but in these simulations the North Atlantic is even colder than the modern ocean. The multiple optimization Earth System model of Mid-Miocene based on cGENIE required CO₂ concentrations as high as 1120 ppm to achieve the highest North Atlantic temperatures without overestimating tropical temperatures, simulating a latitudinal thermal gradient of 15.8 °C⁴⁷. In the case of the Mid-Miocene there is one recent water isotope enabled Community Earth System Model output (MIO400 CI; 400 ppm of CO₂) that publicly includes mean annual water column temperatures⁶. Surficial mean annual temperatures from this model (16.2 °C) only differ in 0.9 °C to mean annual 0-70 m (modern euphotic zone limit for Site 982) temperatures (15.3 °C), matching our coccolith Δ_{47} calcification temperatures for the Mid-Miocene. While the gradients between EEP alkenone-derived temperatures and our North Atlantic coccolith Δ_{47} winter-spring and mean annual temperature records are more similar to models^{4,6,47} (~11.2, and 9.9 °C, respectively), the gradient suggested by other published proxies is negligible (2.7 °C) (Fig. 5, Supplementary Fig. 9). This analysis shows that coccolith Δ_{47} calcification temperatures are more consistent with modeled Late and Mid Miocene latitude thermal gradients and suggest a lower degree of high latitude amplification for the North Atlantic than that inferred from other proxy data.”

The new supplementary note no reads:

“Supplementary Note 5. Coccolith Δ_{47} calcification temperatures comparison to modelled mean annual temperatures

Traditionally, studies focused on model-data comparisons take directly proxy estimates and compare them to modelled data, without considering or correcting for potential biases of any kind. A typical case is when alkenone derived temperatures for the North Atlantic estimated using the Bayspline alkenone calibration²⁴ are used for direct comparisons with modelled outputs, as the proxy data will be biased towards August-October temperatures (warm bias). Despite this, no efforts to “back-calculate” the seasonally-biased proxy to mean annual temperatures are traditionally conducted in these studies, nor are models comparing their seasonal outputs to proxies, but rather they use mean annual estimates directly.

Ideally, we would compare seasonally-modelled data at depth to our coccolith Δ_{47} calcification temperatures. However, due to the lack of publicly available winter-spring temperatures and the water column temperature structure from the majority of models⁴⁰, we have had to take the traditional approach and compare our coccolith Δ_{47} calcification temperatures directly to model mean annual temperatures. Then only exception is the recently published model MIO400 CI which does include mean annual data of the water column for our study site⁴² for the Mid-Miocene, allowing a more direct comparison to our coccolith Δ_{47} calcification temperatures. Although potential effects of seasonality cannot be isolated from the data of this model, both mean annual SSTs and mean annual 0-70 m (modern euphotic zone) temperatures match our coccolith Δ_{47} calcification temperatures with a difference of only 0.9 °C between depths.

In an attempt to account for potential seasonal and depth of production cold bias, we have as well back-calculated surface mean annual temperatures from our coccolith Δ_{47} calcification temperatures, assuming the difference between surface mean annual temperatures and winter-spring temperatures at production depth throughout the Miocene is similar to that of the modern ocean. We highlight that this assumption is likely not valid in a changing ocean since the Miocene, since we expect variability in for instance, the temperature contrast between seasons, coccolithophore season and depth of production depending on nutrient and mixing strength, and in the relationship between SSTs and temperatures at depth at our site.

To back-calculate the seasonal effect (~ 0.97 °C) we used temperatures from the season of production suggested by Behrenfeld et al.²⁰ (December-June) and compared them to mean annual SSTs for our site (10.57 °C). For the depth of production effect (~ 0.34 °C), we used the average SST for December-June (season of production²⁰) and the average temperature of the water column between 0 and 70 m, considering this is a well-mixed site and that 71 m is the limit of the euphotic zone layer below which light during the season of production is not sufficient for photosynthesis. We calculate that assuming the ocean since the Miocene was as it is now, we could have a cold bias effect of 1.31 °C, which we have added to calculate surficial mean annual temperatures and compare them to modelled mean annual temperatures (Figure 5, Supplementary Figure 9).

It is worth noting that the comparatively larger maximum seasonal effect mentioned for the Holocene sample in the main text (up to 3 °C) is obtained when instead of using mean annual temperatures for calculations, SSTs of August to October (used in the Bayspline calibration²⁴; warmer than mean annual temperatures) are considered, and the season of production proposed by Broerse et al.¹ (March to May) is used instead of the broader December to June suggested by Behrenfeld et al.²⁰. Regarding depth of production, for the Holocene sample maximum effect we used the maximum depth bias which is obtained for June for a 100 m depth (Supplementary Table 3; ~ 1.6 °C), instead of considering the average temperatures of the water column until the limit of the euphotic zone during months of production.”

There are also a couple of instances where there is an opportunity to compare to existing data. These are explained below, along with minor comments relating to specific lines of the manuscript.

2. Lines 174-177: How are the SSTs during season of production known for the mid-Holocene?

Here we assume that the modern North Atlantic oceanographic and ecological conditions for coccolithophores represent well the North Atlantic during the mid-Holocene in terms of season and depth of production. We have now clarified this in the main text:

“...For the Mid-Holocene ODP Site 982, published alkenone-derived temperatures using the core-top²⁰ and Bayspline²¹ calibrations were up to 5.9 °C warmer than both modern SSTs during the season of production and coccolith Δ_{47} calcification temperatures¹⁷. Assuming the modern North Atlantic represents well the Mid-Holocene conditions at the same site, the maximum effects of applying alkenone calibrations based on 1) SSTs rather than on temperatures at depth of production and 2) SSTs during a warmer season than that of production, can explain up to 78% of the difference in published Mid-Holocene absolute temperature estimates between coccolith Δ_{47} and alkenones¹⁷...”

3. Lines 206-208: How about the Herbert et al. (2016) Uk37 data? Here and in Fig. 4.

We have included the comparison with data of Herbert et al., (2016) in Figure 5 (reference 1). We have deleted the data from Herbert from other main figures because it was previously asked by other reviewers to simplify them because they were too “crowded”, especially because Herbert’s data overall agrees well with our alkenone data (same trends, within the confidence interval of the Bayspline calibration for our data). We therefore think we should keep figure 4 without including Herbert’s data (but including it in fig 5), following the suggestion by previous reviewers, but we have now directly made this comparison in the main text, which now reads:

“...Applying this calibration to our ODP 982 $U_{37}^{k'}$ values and those previously published for our site¹, we obtain absolute alkenone-derived growth temperatures that agree much better with the absolute values of our coccolith Δ_{47} record. The same is true when this calibration is applied to the higher resolution ODP Site 982 $U_{37}^{k'}$ Miocene values of the study of Super et al.⁸, which decreases average alkenone temperatures by ~ 6.6 °C (Fig. 4)...”

This is how Figure 4 looks like including only the original Herbert’s data (in yellow, reference 1). Including the alternative dataset using the Mejia et al., 2023 calibration would make it even more dense to read.

4. Lines 265-266: Leutert et al. (2020) and Hou et al. (2023, Climate of the Past) have Miocene D47 records from the Southern Ocean, which I think should be included in the discussion of meridional temperature gradients and Figure 5. Though they are in the Pacific sector and not the Atlantic, the authors choose an EEP site for their equatorial temperatures so it seems like this discussion is global.

First, we would like to highlight that with this paper, we have not attempted to generalize a polar scale conclusion of more modest polar amplification. We have stated in several parts of the manuscript that this conclusion is only valid for the North Atlantic, and more data would be needed to prove this in other high latitudes and perhaps in other warm intervals. We have chosen to compare our North Atlantic temperatures to EEP temperatures, as it is a widely-used record for tropical latitudes (Herbert et al., 2016). However, we could have used records from other tropical latitudes and the comparison would still be valid, because the tropical temperatures are not within the scope of discussion of this paper, and they would stay constant for comparisons. We have specified in the main text that a second source of overestimates of high latitude amplification could be underestimates of tropical temperatures by alkenones (e.g. due to oversaturation of the proxy). But again, even if this was the case, comparison to our North Atlantic coccolith clumped record would still lead to more modest amplification than comparisons with North Atlantic alkenone records.

This being clarified, we argue that with this paper we are focusing on the new application of clumped isotopes to coccolith calcite. As opposite to foraminifers, applying clumped isotopes to coccoliths ensures that the source of the signal is from euphotic oceans, which makes its application a better match for surficial water column temperatures. Moreover, the tight control of biomineralization inside the cell to build chemically homogeneous single coccoliths and their cover in polysaccharides make them less prone to dissolution/recrystallization effects compared to foraminifera calcite. We have included a detailed analysis of seasonality and depth of production for coccolithophores in our manuscript, and our samples are enriched in alkenone producing species, which also ensures the best comparison to alkenone temperatures. On the other hand, the study of Leutert et al., 2020 uses clumped temperatures from planktonic foraminifera (*G. bulloides*), for which coccolithophore's seasonality and depth of production may or not apply. Comparison of foraminifera-derived clumped isotope temperatures to coccolith would require conducting assumptions on the foraminifera habitat depth and seasonal variation compared to coccolithophores. As we explained with our comparison analyses with TEX records, comparing temperature proxies with a different organism source is not straightforward, as seasonality and depth of production may be different (or non-thermal effects). In the study of Hou et al. (2023), they use benthic foraminifera to reconstruct bottom water temperatures from the Southern Ocean, which makes comparisons to temperatures from the North Atlantic euphotic ocean even more complicated.

Since we would rather keep the story simple, easy to understand and as accurate as possible, we prefer to focus on comparisons to temperature proxies that have been suggested as surface ocean tracers, especially with alkenones because they are produced by the same organism (coccolithophores). Moreover, we rather limit comparisons to our core location in the North Atlantic,

while specifying our conclusions are only valid for this area in high latitudes. Extended records of coccolith clumped isotopes in other sites during warm intervals like the Miocene would provide further information in future studies to validate or contradict our hypothesis of a more modest high latitude amplification globally.

5. Lines 292-293: Can the authors explain why alkenones would represent well temperatures in the EEP? Though the seasonal temperature swings are less at low latitude, this area has a strong seasonal upwelling cycle with changes in productivity, SST, and the depth of the mixed layer. The authors have made a good case for alkenones being affected by seasonality and depth, which I would think would be a problem in the EEP.

The reviewer is correct. In our paper from 2023 (Mejia et al., 2023), coretop data from the EEP (and other tropical regions) suggest alkenones could have a larger warm bias in warmer, stratified oceans than in well mixed ones (like the North Atlantic), when applying traditional calibrations based on SSTs. Since we do not have a downcore record of the EEP, we have to limit the scope of this paper to discuss North Atlantic clumped coccolith temperatures. Please see the response to reviewer e point 4, where we clarify why our conclusion of a more modest polar amplification using clumped isotopes from coccoliths is independent of the actual magnitude of temperature in the tropics. Despite this, we have now included as well a description of the potential for a larger warm bias of alkenones in the tropics in this version of the manuscript, as suggested by the reviewer.

The paragraph in the main text now reads: *“Miocene extreme polar amplification has been best simulated using CO₂ concentrations around the maximum values (or higher) than those suggested by proxies^{4,47,52}, but high latitude warmth in places like the North Atlantic (and the high latitude southern hemisphere) continues to fall short in model simulations, while tropical temperatures tend to be overestimated. Our North Atlantic coccolith Δ_{47} temperatures suggest there is an overestimated Miocene high latitude warmth associated to the alkenone proxy interpretation, but extreme amplification could additionally be caused by underestimates in tropical temperatures, mostly derived from alkenones as well⁴. A cold bias in tropical and subtropical regions could arise from the lower $U_{37}^{k'}$ sensitivity to high temperatures⁵³, and the analytical problem to detect C_{37:3} alkenones when $U_{37}^{k'}$ approaches the limit of one⁵⁴. In line with this, a less extreme polar amplification for the Pacific Ocean was reported for the Late Miocene when instead of alkenones, Mg/Ca from a mixed layer foraminifera was used as proxy to reconstruct tropical temperatures⁵². On the other hand, it is also possible that the application of traditional calibrations based on SSTs^{20,21} to alkenone data from tropical, stratified oceans lead to larger overestimates of SSTs compared to more mixed areas like the North Atlantic¹⁷. If this was also valid for the Miocene, it would result in increased polar amplification. Regardless of the magnitude of tropical temperatures, comparisons to our North Atlantic coccolith Δ_{47} temperature record produce a more modest polar amplification than when compared to alkenone-derived temperatures.”*

6. Line 365: I think you mean “at 2 to 10 μm ”?
Modified to “at 2 and 10 μm ”

7. Figure 1: Caption says red star but it is black. I suggest adding a mark indicating the 6.74C modern temperature corresponding to the Icelandic data.
Modified as suggested by the reviewer.

8. Figure 2: The Uk37 all batch culture symbol in the legend is invisible for me.
Modified as suggested by the reviewer.

9. Figure 5: When I think of meridional temperature gradients, I think of Figure 12 from Burls et al. (2021). Since the authors are comparing directly to that model, it would be very compelling to show the alignment of temperatures calculated herein in the context of that type of figure. This is just a suggestion, but I think it would be helpful for the readers and make a more compelling case to show the Burls curves somewhere.

We have now modified the format of presenting Figure 5 by including the curves of the Burls models instead of only the datapoints for the North Atlantic and EEP, as suggested by the reviewer. We have also included now the mean annual SST and mean annual average 0-70 m water column

temperatures from the recent model of Lee et al., 2025 to make a more direct comparison to our clumped isotope data. We have left a simplified version of this figure in the main text, which does not include errors of our proxy data, so as to improve readability (in the figure caption we give written errors). However, we now include a version of the same figure as Suppl. Fig 9, in which we include errors as well.

10. Supplementary Methods oceanographic setting: Since so much of this work relates to seasonality and the authors make a conversion from seasonal to annual temperature, it would be helpful to see some numbers on the seasonal swings in SST and even euphotic zone temperature, if available.

The seasonal changes of SSTs are shown already both in the Supplementary Tables 3 and 4 of this manuscript. A more detailed analyses of variations of seasons in the modern ODP Site 982 can be found in the Supplementary information of Mejia et al., 2023. If the reviewer considers it necessary, we could add some form of figures from the Mejia et al., 2023 Supplement to the supplement of this manuscript. For instance: 1) SST and mixed layer variations over the year (see below, left. Green bar denotes months of production), and 2) vertical temperature profile (purple), photosynthetic active radiation (PAR, dashed light blue), PAR values between 10-1% (blue horizontal lines), base of the mixed layer during production months (green shade), coccolith clumped isotope temperatures and suggested depth of production (triangle and circle, different calibrations). Please let us know if you consider this necessary.

We thank the reviewers for their constructive comments, which allowed us to improve our manuscript. Please find below our response to specific comments in blue, and in *blue italics*, the extracts of the manuscript text that relate to those comments

REVIEWER COMMENTS

Reviewer #1 (Remarks to the Author):

I would like to thank the authors for their detailed responses to my comments on the previous version. I find the manuscript improved following the revisions, and I am pleased to recommend publication pending the following minor editorial revisions.

1. L110: Suggested revision of subheading “Assessing the fidelity of coccolith Δ_{47} temperatures: negligible evidence for a cold bias”

Done, as suggested by reviewer. But when looking at editorial checks we realized we have to delete this subheading, so we made it part of the first paragraph.

“When assessing the fidelity of coccolith Δ_{47} temperatures, several lines of evidence suggest that coccolith Δ_{47} reflect the primary calcification temperature of coccoliths with negligible influence from variable vital effects, diagenetic overprinting, nor cold biases.”

2. L163–165: Consider citing a more comprehensive review on UK’37 calibration (including cultures) by Herbert, 2001 G-cubed (<https://doi.org/10.1029/2000GC000055>)

Included.

3. L350: Bayspline  BAYSPLINE

Changed in the whole document and in the supplement

4. L384: “...publicly includes...” reads awkward. Please rephrase for clarity.

We rephrased the sentence as:

“In the case of the Mid-Miocene there is one recent water isotope enabled Community Earth System Model output (MIO400 CI; 400 ppm of CO₂) that includes open access to mean annual water column temperatures⁶.”

5. L386: Suggested revision “...differ by 0.9 °C from the mean annual temperature of the upper 0–70 m water column...”

Modified, as suggested by reviewer

6. L482: PoraPak  Porapak (this is a registered trademark)

Modified, as suggested by reviewer

Finally, I would like to congratulate the authors on this excellent piece of work, which provides valuable insights and challenges our current understanding of the widely applied UK’37 proxy and the latitudinal temperature gradient during the warm Miocene.

Reviewer #2 (Remarks to the Author):

I am satisfied with the authors’ responses. Since, as the authors point out, this manuscript is more focused on the potential of the clumped isotope proxy, and that is not my area of expertise, I leave that to Reviewer 1. Below are a few minor things I noticed from the parts of the manuscript related to my review.

1. Figure 4: The “a” label is a bit confusing, perhaps move it either up to the top of the box or down to the top of the data.

Moved to the top of the data.

2. Line 303: I'm not sure what you mean by magnitude here, a few more words are needed.

We changed "magnitude" by "absolute values"

3. Line 325: "could be invalid"

Modified, as suggested by reviewer

4. Line 759: "from which it differs by"

Modified, as suggested by reviewer

5. Line 760: "between 4 and 4.5" otherwise use a dash after 2.8

Modified, as suggested by reviewer: "between 4 and 4.5"

6. Supplement line 327: "The only"

Modified, as suggested by reviewer